# Pellioditis pelhamensis n. sp. (Nematoda: Rhabditidae) and Pellioditis pellio (Schneider, 1866), earthworm associates from different subclades within Pellioditis (syn. Phasmarhabditis Andrássy, 1976)

**Irma Tandingan De Ley[1], Karin Kiontke[2], Wim Bert[3], Walter Sudhaus[4], David H. A. Fitch[2]***

1 Department of Nematology, University of California, Riverside, CA, United States of America,
2 Department of Biology, New York University, New York, NY, United States of America, 3 Nematology Unit, Department of Biology, Ghent University, Ghent, Belgium, 4 Institut für Biologie/Zoologie, Freie Universität Berlin, Berlin, Germany

* david.fitch@nyu.edu

**Data Availability Statement:** Sequence files are available from the GenBank database (accession

## Abstract

Recently, much attention has been focused on a group of rhabditid nematodes called *Phasmarhabditis*, a junior synonym of *Pellioditis*, as a promising source of biocontrol agents for invasive slugs. *Pellioditis pelhamensis* n. sp. was first isolated from earthworms near Pelham Bay Park in Bronx, New York, USA, in 1990 and has been found to be pathogenic to slugs as well as some earthworms. It has also been used in several comparative developmental studies. Here, we provide a description of this species, as well as a redescription of a similar earthworm-associated nematode, *Pellioditis pellio* Schneider, 1866, re-isolated from the type locality. Although *P. pelhamensis* n. sp. and *P. pellio* are morphologically similar, they are reproductively isolated. Molecular phylogenetic analysis places both species in a clade that includes all species previously described as *Phasmarhabditis* which are associated with gastropods. *Phasmarhabditis* Andrássy, 1976 is therefore a junior synonym of *Pellioditis* Dougherty, 1953. Also, *Pellioditis bohemica* Nermuť, Půža, Mekete & Mráček, 2017, described to be a facultative parasite of slugs, is found to be a junior synonym of *Pellioditis pellio* (Schneider, 1866), adding to evidence that *P. pellio* is associated with both slugs and earthworms. The earthworm-associated species *P. pelhamensis* n. sp. and *P. pellio* represent different subclades within *Pellioditis*, suggesting that *Pellioditis* species in general have a broader host range than just slugs. Because of this, caution is warranted in using these species as biological control agents until more is understood about their ecology.

## Introduction

Combatting pest organisms with biological means is an attractive alternative to pest control with chemicals. However, biological control is not without risks. In the past, several species

numbers OR059186 and OR059187). All other primary data are within the manuscript and its Supporting Information files.

**Funding:** This study was supported by National Institutes of Health grant R01GM141395 to DHAF and funds provided to ITDL from the United States Department of Agriculture Specialty Crop Multi-State Program (USDA-SCMP) in partnership with CDFA grant #12509488. The funders had no role in study design, data collection and analysis, decision to publish, or preparation of the manuscript.

**Competing interests:** The authors have declared that no competing interests exist.

meant to serve as specific biocontrol agents had unintended negative consequences for biodiversity (reviewed in [1]). This is especially true if a species used for biological control is not specific for the intended target, or if it colonizes habitats far outside of the range of the initial application.

The continued search for safer alternatives to toxic molluscicides in mitigating invasive gastropod pests led to the successful commercialization of the nematode *Phasmarhabditis hermaphrodita* (a.k.a. *Pellioditis hermaphrodita*) as the biological control agent Nemaslug®. This search also focused interest into related species. Of the 21 species described so far in this group (15 discovered since 2015), all but three were isolated from slugs or snails [2]. However, one species related to *P. hermaphrodita*, represented by strains EM434 and DF5056 and until now unnamed, was isolated from earthworms [3]. Two additional strains, DL308 and DL316 (with identical mitochondrial DNA sequences as EM434) were recently recovered in Oregon from the slugs *Deroceras reticulatum* [4] and *Testacella* sp. (D. Howe & D. Denver, pers. comm.). Thus, this species naturally uses both molluscs and earthworms as natural hosts. Two other related species, *Pellioditis pellio* and *Pellioditis mairei*, were originally described as associates of earthworms, but *P. pellio* (described as *Pellioditis bohemica*, syn. nov.) has also been reported from slugs [5, 6]. Thus, at least some species in this group are not specific to mollusc hosts.

Recent experiments showed that EM434/DF5056 causes mortality to the agriculturally important slug pests *D. reticulatum* and *Ambigolimax valentianus* and to the earthworm species *Eisenia hortensis* and *Eisenia fetida* (composting worms), but not to *Lumbricus* [7]. Thus, even though the natural association of these species with their host appears to be "necromenic", they are pathogenic under experimental conditions or perhaps in a non-native host. Necromeny ("waiting for the cadaver") describes a lifestyle in which nematode dauer larvae enter a host but do not resume development until after the host dies [8]. Subsequently, they exit the dauer stage and develop on the decaying cadaver. Necromenic nematodes are normally not harmful to the host; e.g. earthworms may encapsulate them in "brown bodies" [9].

To avoid unintentional effects on the earthworm population should further *Pellioditis* (syn. *Phasmarhabditis*) species be developed as biological control agents, the host range of these species needs to be more fully known. As a beginning, it is worthwhile to ask if the earthworm association is an evolutionary novelty within the group or an ancestral feature of the group. Knowledge of the phylogenetic relationships of the species in the taxon can help decide between these possibilities, but it requires molecular data for the earthworm-associated species, and thus isolation of a reliable strain of *P. pellio*.

Reconstructing the phylogeny of *Pellioditis* including *P. pellio* should also settle a taxonomic controversy involving the genus [2, 10]. In 2011, Sudhaus [11] declared *Phasmarhabditis* to be a junior synonym of *Pellioditis*. This was because *P. pellio*, the type species of *Pellioditis*, was recognized as part of the taxon based on morphological similarities. Once a type species is transferred into a different genus (here into *Phasmarhabditis*), its genus name (here *Pellioditis*) has priority if it is an older name; the junior synonym should then be retired (see S1 File; ICZN Articles 23 and 61 [12]). This synonymization was not generally accepted by researchers working on mollusc-pathogenic nematodes who continued to use the name *Phasmarhabditis*. They argued either that the preferred genus name was a matter of "diverse perspectives of revisions" [13, 14], or that *Phasmarhabditis* and *Pellioditis* were "clearly separated phylogenetically" [6, 15, 16]. The latter argument was derived from a mistaken taxon sampling in molecular phylogenetic analyses, where the genus *Pellioditis* was represented by two species, *Litoditis marina* and *Litoditis mediterranea*, that were listed as *Pellioditis* species in GenBank (now corrected). These species had been removed earlier from *Pellioditis* [11] and transferred into genus *Litoditis* after a molecular phylogenetic analysis of rhabditid nematodes revealed that the genus was previously polyphyletic [17]. Because this analysis did not include *P. pellio*, and indeed, no

sequences for *P. pellio* were yet available in databases, *Phasmarhabditis* researchers might be forgiven for their confusion over the taxonomy (see S1 File for a full history of *Pellioditis* and *Phasmarhabditis* taxonomy and arguments for synonymy).

Here, we provide a description of the earthworm-associated species represented by strain EM434 as *Pellioditis pelhamensis* n. sp. and a redescription of *P. pellio* based on a new isolate collected from the type location and type host. We reconstructed a phylogeny using molecular sequence data we obtained for *P. pellio* and *P. pelhamensis* along with all available sequences of species described as *Phasmarhabditis* as well as representatives of the outgroup and of the two mollusc-parasitic clades *Agfa* and Angiostomatidae. This analysis shows clearly that *P. pellio* is part of a monophyletic clade that includes all *Pellioditis/Phasmarhabditis* species plus Angiostomatidae. *P. pelhamensis* and *P. pellio* belong to two different subclades of *Pellioditis*, indicating that earthworms were either acquired as hosts twice independently, or that an earthworm association is an ancestral feature of *Pellioditis*.

## Materials and methods

### Nematode isolation and culture

Strains EM434 and DF5056 were isolated from earthworms, which were first washed in tap water and then chopped up on a nematode growth medium (NGM) agar plate. Rhabditids that came out were collected over the next few days. One individual gravid female was picked to start each strain. Cultures were maintained on *Escherichia coli* (OP50). Strains were cryogenically maintained shortly after recovery. EM434 was used for the description reported here.

*P. pellio* strain SB361 was isolated by WS from the type location for this species [18]. Pieces of earthworm were placed directly on water agar and emerging nematodes were recovered. A culture was provided to the Fitch lab and cryogenically preserved on June 17, 2005, by KK. SB361 was thawed in 2022, grown on NGM with *E. coli*, and used for the redescription reported here.

### Identification of nematodes and crosses

That strains EM434 and DF5056 belong to the same species was shown by interfertility and essentially identical nucleotide sequences for SSU and LSU (small and large subunit ribosomal RNA genes) and CO1 (cytochrome oxidase I gene) [7]. Detailed morphological characterization was thus performed on only one strain, EM434, as the representative strain for the species. EM434 was previously suspected to be *P. pellio* based on morphology, so rRNA (ribosomal RNA) genes including the ITS (internal transcribed spacer) regions were sequenced from strains SB361 and EM434 (see S1 Table). Ribosomal RNA gene sequences have proven to be good for phylogenetic analysis of rhabditids and the ITS sequences for barcodes [17, 19–23].

Measurements were obtained from nematodes fixed in double-strength formalin-glycerol and processed to anhydrous glycerin for light microscopy as previously described [24, 25]. Some measurements were also obtained from live nematodes anaesthetized with sodium azide and not found to be significantly different. Line drawings were made from live, anaesthetized nematodes using a Zeiss Axioplan microscope equipped with a drawing tube and DIC (differential interference contrast) optics.

To test if strains EM434 and SB361 were interfertile or reproductively isolated (i.e. conspecific or not according to the Biological Species Concept), we performed reciprocal interstrain and intrastrain control crosses. Three or four females from one strain were left uncrossed (as a control to test their virginity), crossed with three or four males from the same strain (as a control for positive cross), or crossed with three or four males from the other strain (duplicates

were set for these latter crosses). Late J4 or early adult individuals were used to set the crosses on NGM plates. Plates were kept at 20˚C and scored for progeny 4 days later.

## Scanning electron microscopy (SEM)

SEM was performed at the University of Ghent, Belgium as described previously [26] with minor modifications. Specifically, nematodes were fixed in 4:1 formalin-glycerol, rinsed three times with PBS at 10 minute-intervals, followed by another two rinses of distilled water at 10 minute intervals and sonication for 8 min. Nematodes, in as little liquid as possible, were placed inside a metal cylinder lined with fine mesh. This was followed by a dehydration series in increasing concentrations of ethanol, starting at 30% for 15 min., transferred successively to 50, 65, 75, 85, 95 and 98% each for 20 min. and three times at 100% for 10 min. Dehydrated nematodes were critical-point-dried using a Balzers Union dryer (CPD020) mounted on the surface of double-sided conductive carbon tape on a stub, sputter-coated with 25 nm layer gold (Jeol, SOP Renaat Dasseville) for 3 min., and observed on a JEOL JSM-840 scanning electron microscope at 5 kV.

## Immunofluorescent staining

Adherens junctions in EM434 were stained using MH27 monoclonal antibody as detailed previously [27]. The patterns of junctions in the mid-J4 males were used to assign homologies of ray (genital papilla, GP) cells with those of other rhabditids [23, 27, 28] using GP nomenclature described previously [29].

## Molecular phylogenetic analyses

DNA sequencing of the entire ribosomal RNA (rRNA) gene repeat unit (18S or SSU rRNA, 28S or LSU rRNA, the 5.8S rRNA gene and flanking internal transcribed spacers ITS1 and ITS2) was performed as previously described [17]. GenBank accession numbers are OR059186 (nearly complete ribosomal RNA gene repeat unit for *P. pelhamensis* n. sp. strain EM434) and OR059187 (nearly complete ribosomal RNA gene repeat unit for *P. pellio* strain SB361).

Corresponding sequences of representative rhabditid nematode taxa were downloaded from GenBank (S1 Table) and preliminary alignments were obtained using Clustal Omega [30] with default parameters. A supermatrix was constructed by concatenating the alignments of the nuclear SSU, ITS1, 5.8S, ITS2 and LSU rRNA genes, RNAP2 (RNA polymerase II large subunit) coding regions (introns were deleted), and mitochondrial CO1, 16S and 12S rRNA genes, with flanking portions of the mitochondrial genomes. Where available, nuclear and mitochondrial genome sequences were used to scaffold the supermatrix. Manual alignment using Mesquite (ver. 4.70 build 940) [31] was used to optimize the alignment in several regions. For example, for protein-coding loci, inferred amino acid sequences were used with the appropriate nuclear or mitochondrial genetic code to align the cognate nucleotide sequences. Also, the reverse complement of several of the sequences had to be used, since GenBank accessions for these sequences are not uniform with respect to sequence orientation (see S1 Table). BBEdit (ver. 14.6.4) was also used for editing of sequences, alignments and NEXUS commands.

A major problem was that each species is represented by multiple, often different and noncontiguous or nonoverlapping fragments of DNA sequences. Having each sequence represent a separate operational taxonomic unit (OTU) generates a very sparse data matrix with many OTUs and a correspondingly enormous treespace that would be difficult to sample for heuristic searches. We thus decided to reduce the number of OTUs while maintaining as much sequence coverage as possible for each species taxon. To do this, we collapsed several sequences representing the same species into a single OTU by (1) concatenating sequences for

different genes from the same species into a supermatrix, even if they derived from different strains, and (2) reconstructing a most parsimonious ancestral sequence for four cases in which the same gene was represented by multiple sequences from different strains/isolates: *P. hermaphrodita*, *Pellioditis zhejiangensis*, *Pellioditis papillosa* (keeping separate strain ITD510), and *Pellioditis californica* (see S1 Table). The ancestral reconstructions did not involve many sites and was performed manually using other closely related species as outgroup representatives; conventional ambiguity codes were used where a sequence or ancestral state could not be unambiguously determined. Although this produced a supermatrix with a single OTU for most species, four species were still represented by 2–3 OTUs: *Pellioditis bohemica*, *Pellioditis apuliae*, *P. papillosa* and *Pellioditis neopapillosa*. The final supermatrix comprised 13,522 nucleotide characters (alignment positions) and 53 OTUs representing 48 species (NEXUS file available from DHAF on request).

To test robustness to phylogenetic inference method, the supermatrix was analyzed using weighted parsimony (WP) and maximum likelihood (ML) approaches. For WP, the supermatrix was reduced to only the phylogenetically-informative characters (4,578 nucleotide characters) for the 48 species (53 OTUs). Because parsimony does not require an evolutionary model, it is predicted to be quite robust to missing data given a fair representation of characters [32]. WP analysis was performed with PAUP* [33] using a stepmatrix that weighted transversions twice that of transitions, i.e. "weighted parsimony", demonstrated to be significantly more accurate than unweighted parsimony [34]. Several heuristic searches were conducted, using random addition of sequences to get a starting tree (addseq = random), with 1000 replications (nreps = 1000) and the default TBR search method. In each of three separate searches, the same set of 27 equally most-parsimonious (MP) trees were found. In the largest search, approximately $4 \times 10^8$ trees were evaluated. To assess the robustness of the data with respect to phylogenetic bipartitions, jackknife analysis was performed (100 replicates with 25% random character deletion, each replicate employing a heuristic search with the following parameters: addseq = random, swap = tbr, nreps = 10).

For ML, a supertrees approach was used. In this case, a complete set of characters is needed per character set (e.g. gene) to ensure appropriate evolutionary model and parameter estimation. Because the supermatrix was somewhat sparse, the matrix was divided into seven subsets of characters and taxa, minimizing the amount of missing data for each subset (S1 Fig). DatasetA covered a part of the SSU gene and comprised 1,612 characters for 41 OTUs; DatasetB covered ITS and 5.8S genes (717 characters) only for 18 OTUs in the *Pellioditis* clade; DatasetC included the D2/D3 region of the LSU gene (901 characters) for 42 OTUs; DatasetD consisted of nearly complete SSU and LSU sequences (4,963 characters) for two of the *Pellioditis* species (*P. pellio* and *P. pelhamensis* n. sp.) plus 18 outgroup rhabditids; DatasetE included nearly complete SSU, LSU and RNAP2 sequences (6,881 characters) for *P. pelhamensis* n. sp. and 17 outgroup rhabditids; DatasetF only included partial sequences from the CO1 gene (474 characters) for 8 *Pellioditis* OTUs plus 7 outgroup rhabditids; and DatasetG covered partial sequences from the mitochondrial 12S and 16S rRNA genes (1,176 characters) for 10 *Pellioditis* OTUs and 7 outgroup rhabditids. RAxML [35] was used to perform ML bootstrapping analysis individually on each of the seven subsets (1000 bootstrap replications using a GTR substitution model with a gamma correction for rate variation and invariant sites, with parameters estimated from the data). Nodes (bipartitions) supported ≤60% in bootstrap analyses were collapsed and the resulting trees were used in a supertree analysis using Clann ver. 4.2.5 [36]. Using the MPR (matrix representation using parsimony) criterion, a heuristic search (parameters: TBR branch-swapping, addseq = simple, swap = tbr, nreps = 100) of >$3 \times 10^6$ supertrees yielded 133,302 MPR trees for which a majority-rule consensus was constructed; nodes represented in ≤70% of the supertrees were collapsed to represent uncertainty.

## Nomenclatural acts

The electronic edition of this article conforms to the requirements of the amended International Code of Zoological Nomenclature (ICZN), and hence the new names contained herein are available under that Code from the electronic edition of this article. This published work and the nomenclatural acts it contains have been registered in ZooBank, the online registration system for the ICZN. The ZooBank LSIDs (Life Science Identifiers) can be resolved and the associated information viewed through any standard web browser by appending the LSID to the prefix http://zoobank.org/. The LSID for this publication is: urn:lsid:zoobank.org:pub: F50C9C74-AD2F-4BEC-A147-9B332EB91580. The electronic edition of this work was published in a journal with an ISSN, and has been archived and is available from the following digital repositories: bioRxiv, LOCKSS, PubMedCentral.

# Results and discussion

## *Pellioditis pelhamensis* Tandingan De Ley, Kiontke, Bert, Sudhaus & Fitch n. sp. ZooBank LSID: urn:lsid:zoobank.org:act:C7EC39A9-3D17-47E8-B9FB-AF676B0DDF29

The species epithet is derived from the historical name of the area in which the species was originally found, Pelham, New York. In 1653, Thomas Pell acquired land including what is now the Town of Pelham, the entire borough of the Bronx, islands including City Island, and land along Long Island Sound north to the Rye border and inland to the Harlem river (pp. 46–47 in [37]). This area was named Pelham in honor of Pelham Burton, Pell's tutor, guardian and benefactor (p. 5 in [37]).

## Strains

- EM434: leg. S. E. Baird, 1990, from an unidentified earthworm, Bronx, New York, USA (40.851009, -73.846283). Synonymous designations of this strain:

  = *Pellioditis* sp. [3]
  = *Pellioditis* sp. DF5039 [38]
  = *Pellioditis* sp. EM434 [39–43]
  = *Phasmarhabditis* sp. EM434 [6, 13, 15–17, 26, 44–55]
  = *Phasmarhabditis* sp. [56, 57], using sequence accession numbers EU195967 and
  EU196008.

- DF5056: leg. D. Fitch, 1993, Bronx, New York, USA (40.852514, -73.788823) from an unidentified earthworm. Nuclear and mitochondrial ribosomal RNA gene sequences identical to those of EM434; interfertile with EM434 [7].

- DL308: from a gray garden slug, *D. reticulatum*, Oregon, USA. Nuclear and mitochondrial ribosomal RNA gene sequences identical to those of EM434 [4].

- DL316 from a shelled slug, *Testacella*, Oregon, USA. Molecular sequences identical to those of EM434 (Dana Howe & Dee Denver, pers. comm.).

**Description.**   Based on strain EM434.

*Illustrations*. Figs 1–4, supplemental videos showing stacks of DIC focal planes through live animals (S1–S9 Videos in S2 File).

*Measurements*. See Table 1.

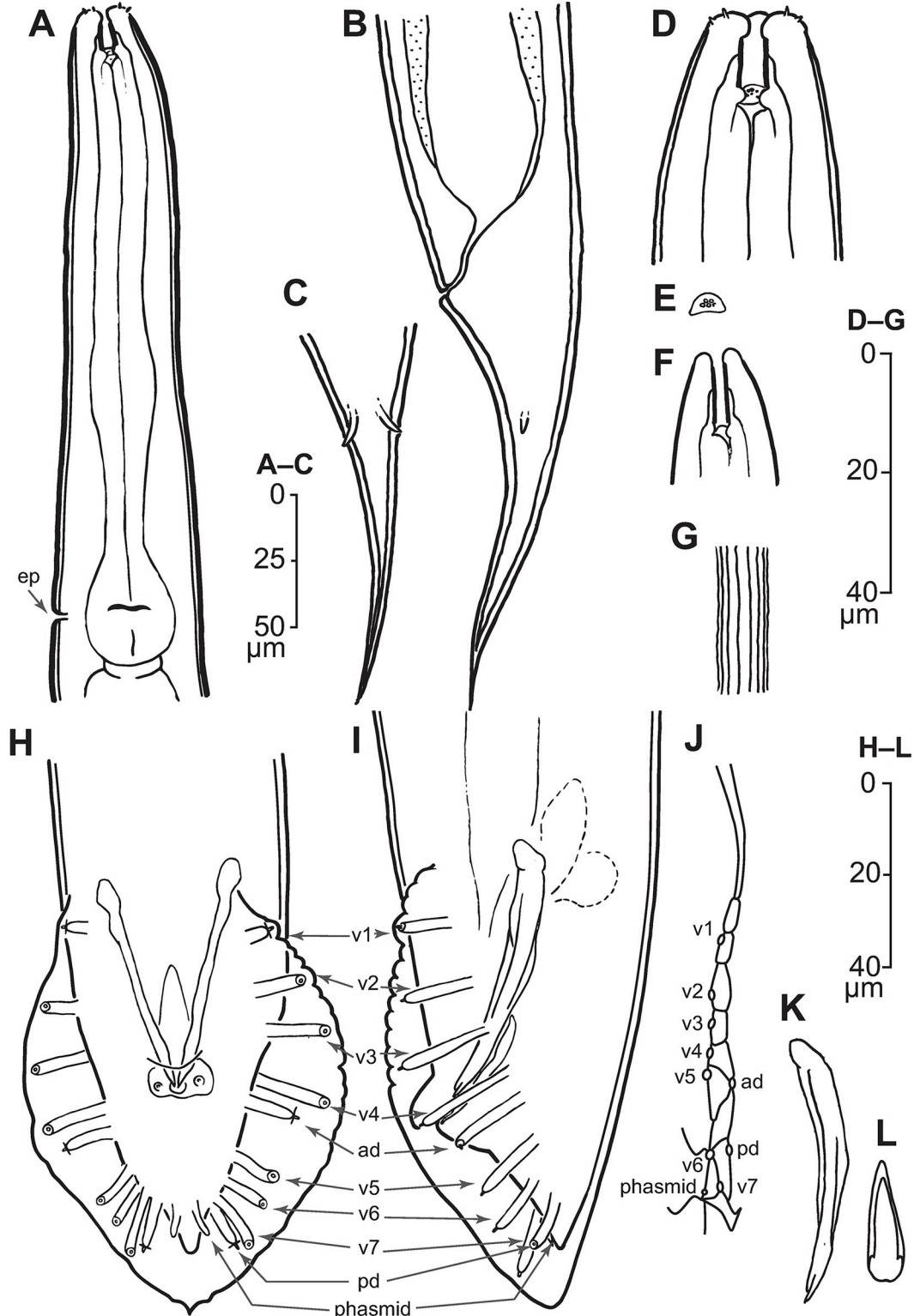

**Fig 1. Line drawings of *Pellioditis pelhamensis* n. sp., strain EM434.** (A–E, G) Adult female: (A) anterior region, left side view, excretory pore (ep) at the level of the grinder. (B, C) Tail region: (B) left side view, (C) ventral view (compare with Fig 2F), showing prominent phasmids approximately halfway down the tail from the anus, protruding through the thick cuticle and terminating on or slightly outside of the external surface. (D) Anterior end showing stoma region, ventral view (compare with Fig 2B showing a left side view). (E) Metarhabdion showing tiny wart-like protrusions. (F) Stoma region of

J2, left side view, showing that the stoma is relatively longer and narrower than in the adult. (G) Lateral field of adult female showing ridges (compare with Fig 2A showing same for male). (H–L) Male: ventral view (H) and left side view (I) of adult male tail with genital papillae (GPs) labeled according to the homology system developed for Rhabditina [28, 29] based on common developmental origins of the GPs in the lateral epidermis [23, 27, 58, 59]. Accordingly, there are two GPs—the anterior dorsal (ad) and posterior dorsal (pd)—with cell lineages originating more dorsally than the seven others (v1–v7) (compare Fig 1J). The small ray-like phasmids (ph) are only seen in ventral view. (J) Adherens junctions of the ray cell and lateral seam cells in the left lateral epidermis of a J4 male (anterior up, dorsal to the right) as visualized by immunostaining with MH27 monoclonal antibody at the time when external ray papillae are forming (compare Fig 2D). (K) Right spicule, left side view; notice the complex tip (compare with Fig 2G). (L) Gubernaculum, ventral view (compare Fig 2G).

*Adults*. Body cylindrical, almost straight to slightly curved ventrad in the middle when relaxed (Fig 3A). Cuticle about 4 μm thick. Annules fine in SEM images, inconspicuous under the light microscope. Lateral field with ca. 5–8 ridges (Figs 1G, 2A, 3G and 4E–4G). Anterior end bluntly rounded, lip region continuous with body. Oral aperture triangular, with slightly convex sides and surrounded by six lips grouped in pairs (Figs 3B–3F and 4B–4D). Inner labial sensilla with protruding dendrite. Outer labial sensilla visible on SEM images, and sometimes in the light microscope as small "bumps" under the cuticle (Figs 3B and 4B). Cephalic sensilla present on dorsal and subventral lips (Figs 3B and 4B). Amphid opening a small inconspicuous slit on a small round mound (Figs 3B–3D and 4B). Stoma short, only slightly longer than lip region is wide. Cheilostom indistinct, gymnostom and pro- and mesostegostom of about the same length (Figs 1D and 2B). Stegostom slightly broader at posterior, ending in well-developed rounded, isotopic metarhabdions, each with several minute tubercles, one central tubercle often more prominent (Figs 1D, 1E and 2B). Stoma wider and shorter in adults than in juveniles (cf. Fig 1F), but with similar proportions between stegostom and gymnostom. Pharynx corpus cylindrical, about two times as long as isthmus with slightly enlarged metacorpus narrowing into isthmus, a bulbous postcorpus (basal bulb) longer than wide, with striated valvular apparatus; isthmus slightly longer than basal bulb (Figs 1A and 2C). Nerve ring encircling isthmus at around 68% of pharynx length. Deirids conspicuous at same level or slightly anterior of excretory pore (Fig 4A). Excretory pore opening at mid-to-base of basal bulb (Fig 1A). Cardia conoid, highly variable.

*Female*. With the characters of adults. Reproductive system didelphic amphidelphic, ovaries reflexed with tips sometimes reaching the level of or past the vulva. Numerous sperm in a long oviduct that has a valve towards ovary and uterus, respectively, forming a spermatheca (Fig 2E). Sperm also found in the uteri. Sperm diameter ca. 10 μm. Uteri of mature females often filled with developing embryos. In older females, hatched juveniles of various stages also present inside of the female uterus. Eggs oval, measuring ca. 50x30 μm. Vulva a transverse slit halfway (50–54%) along the body (Figs 3A and 3G–3I). Vagina length variable, often extending past middle of vulval body diameter. Intestine ending in a rectum of about equal length as anal body width (ABW, Fig 1B). Three rectal gland cells visible. Anus an arcuate slit (Figs 3J–3M). Phasmids prominent, forming small protruding papillae, located at 40 (31–51)% of tail length. Tail conical, with acute tip (Figs 1B, 1C, 2F and 3J–3M).

*Male*. Body habitus similar to female when relaxed, but shorter and narrower. Reproductive system monorchic, occupying 60–70% of total body length; testis ventrally reflexed. Spicules separate with furcated tip, spicule head not set off (Figs 1H, 1I, 1K, 2G and 4H–4K). Gubernaculum spatula-shaped in dorso-ventral view (Figs 1L and 2G). Bursa relatively narrow, anteriorly open, anterior margin slightly wavy (Figs 1H, 1I, 2H–2J and 4H–4K). Nine pairs of genital papillae (GPs, "rays"). Three precloacal GPs. The tip of the first GP (v1) opens at the edge of the fan. The anterior dorsal GP in position 5 (counting from the anterior). The posterior dorsal GP terminal, or originates at the same level as the most posterior GP (v7). The posterior three GPs often form a group. The phasmids form small ventral papillae not fully embedded in the

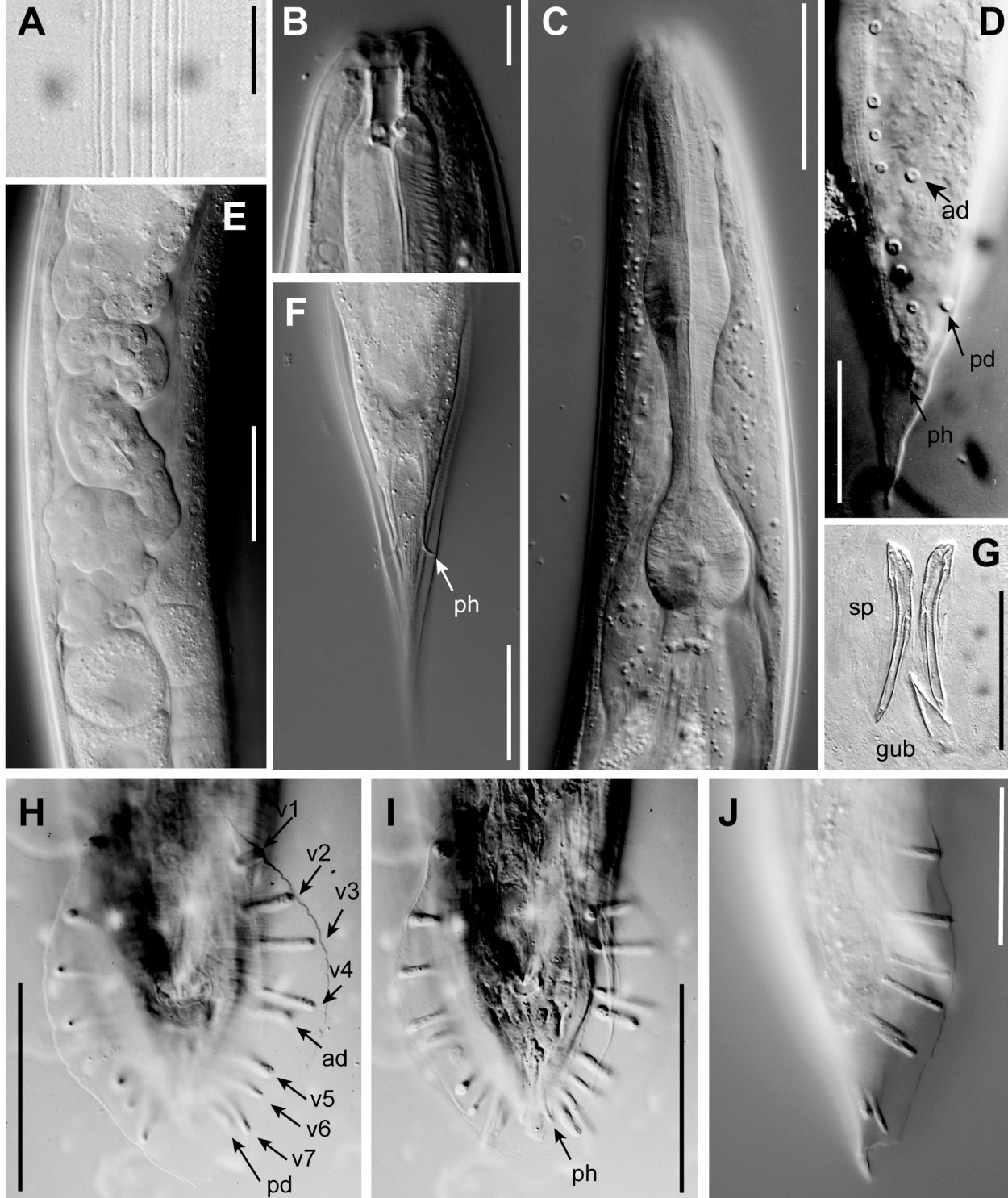

**Fig 2. DIC photomicrographs of *P. pelhamensis* n. sp.** (A) Lateral field of an adult male. (B) Adult male head with stoma, sagittal plane, left side view. (C) Adult male pharynx region, left side view. (D) J4 male tail, left side view, showing developing genital papillae and phasmid (ph) as it is drawn out in the wake of tail tip morphogenesis in this peloderan species; ad and pd indicate the anterior dorsal and posterior dorsal genital papilla (compare Fig 1H–1J). (E) Posterior oviduct and part of posterior uterus of adult female filled with sperm, left side view (anterior up). (F) Adult female tail, ventral view, showing phasmids (ph) projecting through the thick cuticle. (G) Isolated spicules (sp) and gubernaculum (gub) of adult male in ventral view. (H, I) Two different focal planes of the same male tail in ventral view, genital papillae labeled as in Fig 1H–1J, showing a typical GP arrangement (i.e. ad+v4 close together and adcloacal). (J) Male tail in right side view showing variant with ad closer to v5 than to v4. Scale bars 10 μm for A–B, 50 μm for C–J.

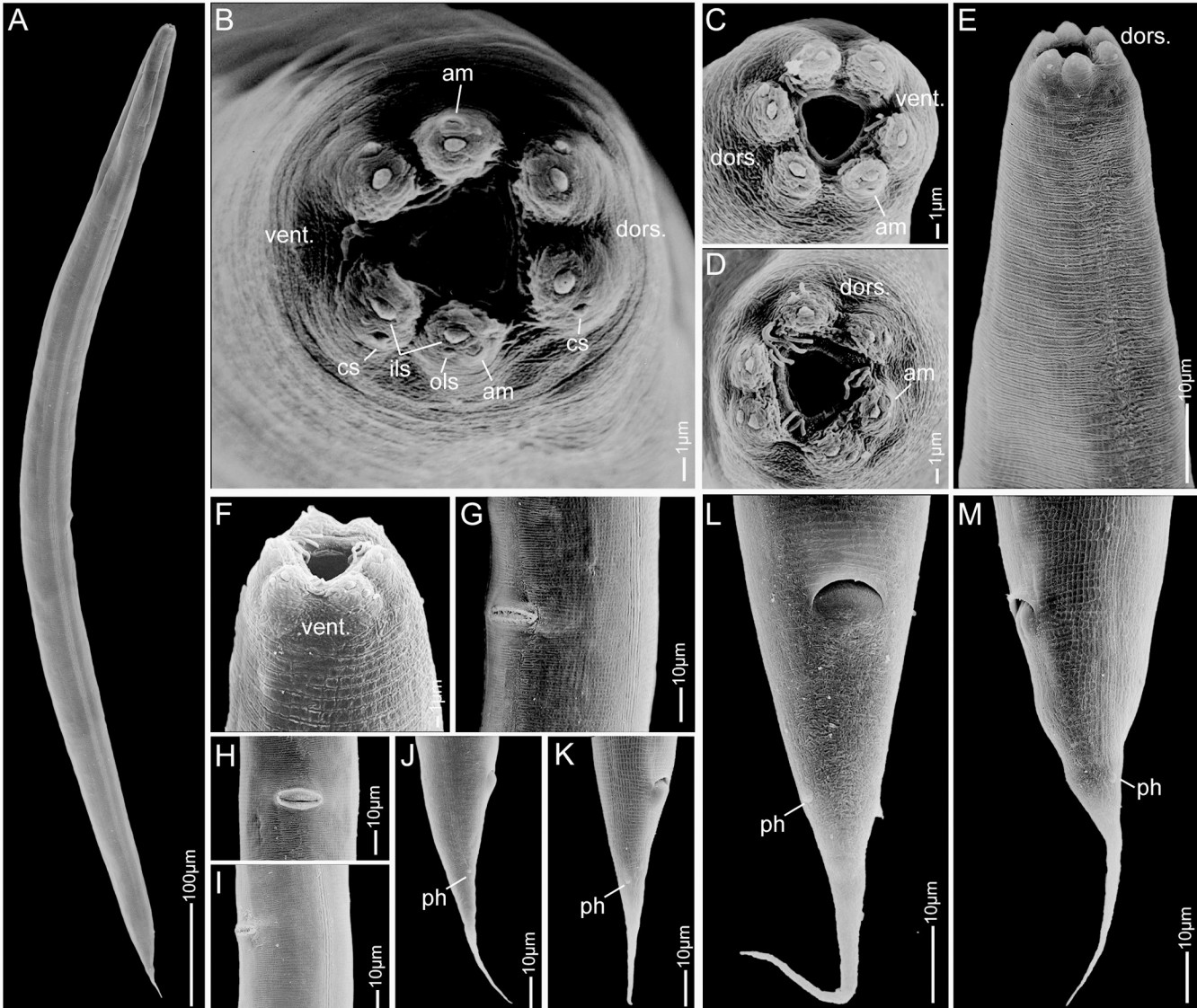

**Fig 3. Scanning electron micrographs of *Pellioditis pelhamensis* n. sp. strain EM434 female.** (A) Habitus. (B–D) Lip region, *en face* view; note that the "bristle-like" features in C and D are bacteria. (E) Anterior region, left side view. (F) Lip region, ventral view. (G) Vulval region with slit-like vulva, sublateral view. (H) Vulva, ventral view. (I) Lateral field, mid-body left-subventral view, vulva at left. (J–M) Posterior region showing conical tail, anus and slightly protruding phasmids, right side views (J, K), ventral view (L) and left side view (M). Abbreviations: dors. = dorsal, vent. = ventral, am = amphid, cs = cephalic sensillum, ils = inner lip sensillum, ols = outer lip sensillum, ph = phasmid.

bursal velum. Thus, the formula (following [11, 29]): v1,v2,v3/(v4,ad)/v5,(v6,v7,pd),ph or v1, v2,v3/(v4,ad)/v5,v6,(v7,pd),ph (where 'ad' and 'pd' represent the anterior and posterior dorsal papillae homologous across rhabditids [23, 27], 'v' represents the other papillae, 'ph' represents the phasmid; here, two slashes '/' represent the adcloacal position of v4+ad). This pattern is also revealed via immunofluorescent staining of the neuroepithelial adherens junctions in J4 juveniles (Fig 1J) at the time when papillae form in the external cuticle (Fig 2D). In strain EM434, variations in the spacing of GP1, GP2 and GP3 observed. Aberrations also found: males with abnormal numbers of rays (one having 13 GPs on one side!) and one adult male with a long (leptoderan) tail tip. Cloacal aperture arcuate slit-shaped, its anterior edge slightly

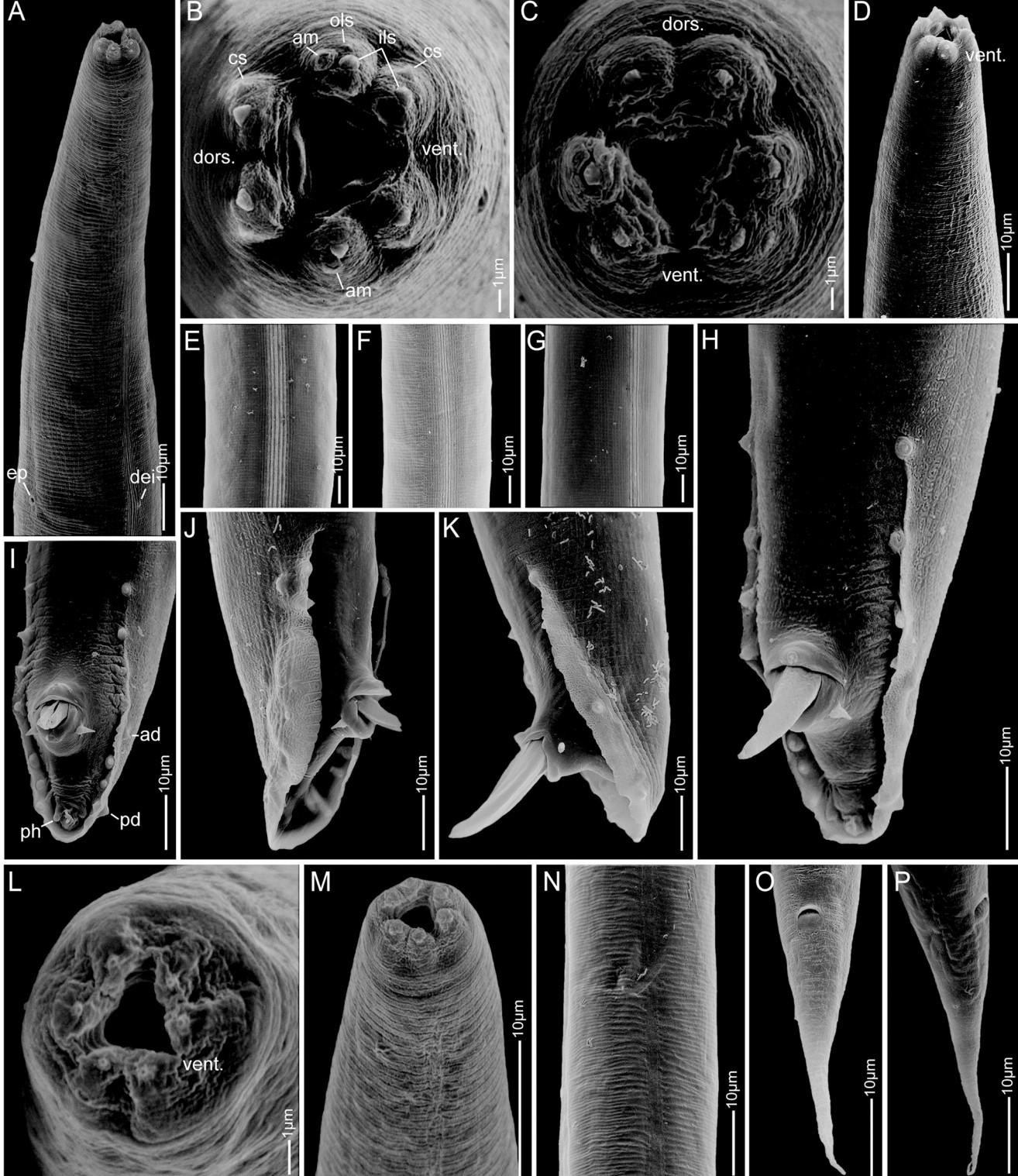

**Fig 4.** Scanning electron micrographs of *Pellioditis pelhamensis* n. sp. strain EM434 adult male (A–K) and juvenile (sex unknown) (L–P). (A) Anterior portion, left-subventral view, showing lateral field; excretory pore (ep) at lower left and deirid (dei) at lower right. (B, C) Anterior end, *en face* view, showing lips separated into pairs: one dorsal and two sublateral. (D) Right-subventral view of anterior. (E–G) Lateral field. (H–K) Male tail, showing bursa, genital papillae, phasmids, spicules protruding from the cloaca, postcloacal sensilla. (H) Left sublateral view. (I) Ventral view. (J) Right lateral view. (K) Left lateral view. (L) *En face* view of juvenile mouth area, dorsal to the upper left. (M) Anterior of juvenile, left side view, dorsal to the right. (N) Lateral field of juvenile.

(O, P) Juvenile tail, ventral and sublateral right side view, respectively, showing anus and slightly protruding phasmids. Scale bars 1 μm in B, C, L; 10 μm in other panels. Abbreviations: dors. = dorsal, vent. = ventral, am = amphid, cs = cephalic sensillum, ils = inner lip sensillum, ols = outer lip sensillum, ad = anterior dorsal genital papilla, pd = posterior dorsal genital papilla, ph = phasmid.

protruding, precloacal sensillum inconspicuous, postcloacal sensilla fairly long with a broad base (Figs 1H and 4H–4K). Tail peloderan.

*Type locality and habitat.* Strain EM434 was started by DHAF in 1990 from a single gravid female from an unidentified earthworm collected by S. Baird on the sidewalk in front of the Forschheimer Building of Albert Einstein College of Medicine (1300 Morris Park Ave., Bronx, NY, USA; GPS coordinates 40.850969, -73.845843). DF5056 was similarly obtained by DHAF in 1993 from an earthworm in lawn soil on City Island (515 Minnieford Ave., Bronx, NY, USA; GPS coordinates 40.8525633, -73.7889098). Neither earthworms were identified at the time, but were similar in appearance to *Lumbricus rubellus*. In conformity with the system used by the *Caenorhabditis* Genetics Center (CGC), these strains were given the strain designations EM434 and DF5056.

*Type material and living cultures.* Of strain EM434, the male holotype (measurements in Table 1) plus another male and two female paratypes (not measured) have been deposited at the Wageningen Nematode Collection, National Plant Protection Organization, Wageningen, The Netherlands, as slide WT3938. The slide contains two females on the left and two males on the right; the rightmost male is the holotype. Another slide containing the paratype represented in Table 1 has been deposited with the Nematode Collection at the University of California, Davis as slide UCDNC 5258. This slide contains six female paratypes; the fourth (counting left-to-right) is the measured paratype. A third slide with two female and two male paratypes has been deposited as slide UGMD_104442 at the Nematode Collection, Ghent University Zoology Museum. Live cultures of EM434 are cryogenically archived at the *Caenorhabditis* Genetics Center (CGC), University of Minnesota, and at the New York University Rhabditid Collection (NYURC).

*DNA sequences deposited at GenBank for strain EM434.* OR059186 (newly reported here): nearly complete repeat unit for the nuclear ribosomal RNA genes (18S, ITS1, 5.8S, ITS2, 28S)

EU161967: RNA polymerase 2 gene, largest subunit, partial CDS

EU196008: 18S ribosomal RNA gene, partial sequence

EU195967: 28S ribosomal RNA gene, partial sequence

MT472231: mitochondrial 16S ribosomal RNA gene, partial sequence, and NADH dehydrogenase subunit 3 (ND3) gene, partial CDS

MT472187: mitochondrial tRNA-Glu gene and 12S ribosomal RNA gene, partial sequence

## *Pellioditis pellio* (Schneider, 1866)

ZooBank LSID: urn:lsid:zoobank.org:act:C1CDBEEF-A1BB-4B0A-AF6A-BC8051416FD8

## Species names synonyms

= *Phasmarhabditis bohemica* Nermuť, Půža, Mekete & Mráček, 2017 syn. nov.
  = *Pellioditis bohemica* (Nermuť, Půža, Mekete & Mráček, 2017) Sudhaus, 2023 syn. nov.

## Strains

SB361: leg. W. Sudhaus, from a *Lumbricus rubellus* Hoffmeister, 1843, found in dry ground in a ruderal area (52.41500, 13.19870) in March, 2004, in Berlin, Germany, corresponding to the

**Table 1. Morphometrics of *Pellioditis pelhamensis* n. sp. strain EM434 and *Pellioditis pellio* strain SB361.**

| Character | *P. pelhamensis* n. sp. EM434 ♀[1,2] | *P. pellio* SB361 ♀ | *P. pelhamensis* n. sp. EM434 ♂[1,2] | *P. pellio* SB361 ♂ | *P. pelhamensis* n. sp. Holotype ♂ | *P. pelhamensis* n. sp. Paratype♀ | *P. pellio* Neotype ♂ | *P. pellio* Topotype ♀ |
|---|---|---|---|---|---|---|---|---|
| n | 15 | 20 | 18 | 15 | | | | |
| L | 1541 ± 122 | 1496 ± 112 | 1155 ± 70 | 1067 ± 57 | 1109 | 1302 | 1149 | 1506 |
| | (1302–1755) | (1302–1698) | (1041–1313) | (974–1155) | | | | |
| a | 13.8 ± 1.1 | 18.7 ± 2.5 | 15.4 ± 1.3 | 20.4 ± 2.1 | 16.0 | 13.1 | 17.6 | 21.0 |
| | (12.3–16.2) | (15.0–25.0) | (12.3–17.5) | (17.2–24.3) | | | | |
| b | 8.0 ± 0.04 | 7.59 ± 0.66 | 6.7 ± 0.6 | 6.0 ± 0.3 | 6.3 | 7.2 | 6.3 | 7.4 |
| | (6.6–7.8) | (6.7–9.0) | (5.9–8.5) | (5.7–6.4) | | | | |
| c | 14.2 ± 1.2 | 13.1 ± 1.2 | 33.5 ± 3.3 | 31.4 ± 2.7 | 31 | 11.5 | 34.8 | 13.8 |
| | (11.5–16.6) | (10.8–15.2) | (25.6–38.3) | (26.6–34.8) | | | | |
| c' | 3.1 ± 0.3 | 4.13 ± 0.5 | 1.0 ± 0.01 | 1.0 ± 0.08 | 1.2 | 3.4 | 1 | 3.9 |
| | (2.7–3.6) | (3.4–5.4) | (0.9–1.4) | (1.0–1.2) | | | | |
| V | 51.9 ± 1.1 | 52.7 ± 0.2 | – | – | – | 51.7 | – | 54.1 |
| | (48.2–52.9) | (45.2–54.4) | – | – | | | | |
| Max. body diam. | 112.7 ± 14.5 | 81.1 ± 9.5 | 75.4 ± 6.5 | 53.0 ± 7.9 | 73.6 | 99 | 65.1 | 71.7 |
| | (94.3–142.4) | (67.9–103.7) | (64.1–90.5) | (42.4–67.0) | | | | |
| Lip region diam. | 12.1 ± 1.7 | 15.4 ± 1.2 | 12.3 ± 1.2 | 13.2 ± 0.7 | 11.3 | 12.3 | 14.1 | 15.1 |
| | (8.5–15.1) | (13.2–17.0) | (11.3–15.1) | (11.3–14.5) | | | | |
| Stoma L | 17.7 ± 1.5 | 21.9 ± 1.6 | 16.5 ± 1.1 | 20.1 ± 0.7 | 18.0 | 18.9 | 20.7 | 22.6 |
| | (14.1–19.8) | (18.9–24.5) | (14.1–17.9) | (18.9–20.8) | | | | |
| Cheilostom L | 5.5 ± 0.4 | 4.7 ± 0.8 | 5.2 ± 0.5 | 4.5 ± 0.7 | 5.7 | 5.7 | 4.7 | 3.8 |
| | (4.7–5.7) | (3.8–5.7) | (4.7–5.7) | (3.8–5.7) | | | | |
| Gymnostom L | 4.0 ± 0.6 | 6.4 ± 0.9 | 3.4 ± 0.6 | 6.0 ± 0.9 | 3.8 | 3.8 | 5.7 | 4.7 |
| | (2.8–4.7) | (4.7–7.5) | (2.8–4.7) | (4.7–7.5) | | | | |
| Stegostom L | 8.2 ± 1.3 | 10.8 ± 2.0 | 8.5± 2.5 | 9.6 ± 0.5 | 8.5 | 9.4 | 10.4 | 14.1 |
| | (5.7–10.4) | (7.5–14.1) | (6.6–17.0) | (8.5–10.4) | | | | |
| Procorpus L | 55.9 ± 4.8 | 59.4 ± 3.9 | 49.2 ± 3.2 | 56.2 ± 3.7 | 48.1 | 49.0 | 55.6 | 61.2 |
| | (49.0–66.0) | (50.0–64.1) | (40.5–53.8) | (51.9–62.2) | | | | |
| Metacorpus L | 29.4 ± 2.9 | 36.7 ± 3.7 | 26.8 ± 4.4 | 31.4 ± 2.1 | 28.3 | 28.3 | 33.0 | 33.0 |
| | (26.4–35.8) | (29.2–43.4) | (21.7–41.5) | (27.4–34.0) | | | | |
| Cardia L | 15.2 ± 4.5 | 8.3 ± 3.1 | 16.3 ± 5.3 | 7.6 ± 1.6 | 20.7 | 15.1 | 10.3 | 4.7 |
| | (7.5–21.7) | (4.7–17.9) | (6.6–24.5) | (4.7–10.4) | | | | |
| Corpus L | 85 ± 7 | 96 ± 5.4 | 76 ± 5.4 | 87 ± 3.8 | 76.4 | 77.3 | 88.6 | 94.3 |
| | (77–101) | (85.8–104.7) | (66.0–88.6) | (79.2–95.2) | | | | |
| Metacorpus diam. | 23.6 ± 2.2 | 22.4 ± 2.2 | 20.9 ± 2.0 | 19 ± 1.3 | 18.9 | 21.7 | 17.9 | 22.6 |
| | (20.7–28.3) | (19.8–27.3) | (17.0–24.5) | (17.9–22.6) | | | | |
| Isthmus L | 48.6 ± 5.1 | 38.9 ± 3.7 | 44.9 ± 5.1 | 35 ± 2.7 | 47.2 | 45.3 | 36.8 | 37.7 |
| | (42.4–58.5) | (30.2–45.3) | (31.1–52.8) | (32.1–39.6) | | | | |
| Basal bulb L | 42.7 ± 2.8 | 40.7 ± 3.7 | 38.3 ± 2.6 | 33 ± 3.2 | 34 | 44.3 | 35.8 | 48.1 |
| | (38.7–49.0) | (33.0–48.1) | (33.9–42.4) | (29.2–37.7) | | | | |
| Basal bulb diam. | 34.0± 2.8 | 30.4 ± 1.6 | 28.0 ± 2.3 | 26 ± 2.0 | 29.2 | 30.2 | 27.3 | 30.2 |
| | (30.2–39.6) | (26.4–33.0) | (21.7–31.1) | (21.7–29.3) | | | | |
| Pharynx L | 193 ± 10.7 | 197 ± 7.4 | 173 ± 9.1 | 176 ± 6.4 | 176.3 | 182.0 | 182.0 | 202.8 |
| | (179–213) | (188–213) | (153–190) | (164–185) | | | | |

(*Continued*)

**Table 1.** (Continued)

| Character | *P. pelhamensis* n. sp. EM434 ♀[1,2] | *P. pellio* SB361 ♀ | *P. pelhamensis* n. sp. EM434 ♂[1,2] | *P. pellio* SB361 ♂ | *P. pelhamensis* n. sp. Holotype ♂ | *P. pelhamensis* n. sp. Paratype ♀ | *P. pellio* Neotype ♂ | *P. pellio* Topotype ♀ |
|---|---|---|---|---|---|---|---|---|
| Nerve ring position | 131 ± 9.7 | 143 ± 8.5 | 116 ± 5.6 | 129 ± 4.5 | 121.7 | 126.4 | 128.2 | 148.1 |
| | (116–154) | (129–159) | (102–126) | (119–137) | | | | |
| E-pore position | 172 ± 14.4 | 182 ± 7.2 | 167 ± 10.8 | 160 ± 8.5 | 162.2 | 160.3 | 174.4 | 190.5 |
| | (159–192) | (171–196) | (149–186) | (148–174) | | | | |
| Vagina L | 60.4 ± 9.5 | 28.6 ± 9.0 | – | – | – | 42.4 | – | 34.9 |
| | (23.3–33.6) | (12.3–44.3) | | | | | | |
| G1 (% G) | 52.8 ± 0.03 | 59.9 ± 0.1 | – | – | – | 47.8 | – | 60.8 |
| | (47.8–56.0) | (50.0–76.3) | | | | | | |
| G2 (% G) | 47.2 ± 0.03 | 41.0 ± 0.04 | – | – | – | 52.2 | – | 39.2 |
| | (44.0–52.2) | (35.0–50.0) | | | | | | |
| Rectum L | 34.4 ± 2.6 | 28.5 ± 4.1 | – | – | – | 33.0 | – | 30.2 |
| | (28.3–37.7) | (21.7–34.9) | | | | | | |
| Anal body diam. | 35.7 ± 3.9 | 28.0 ± 3.1 | 33.9 ± 2.6 | 32.7 ± 2.3 | 30.2 | 33.0 | 33.0 | 28.3 |
| (ABD) | (28.3–42.4) | (20.7–31.1) | (29.2–36.8) | (28.3–35.8) | | | | |
| Anus to phasmid | 40.1 ± 6.5 | 34.4± 4.9 | – | – | – | 38.7 | – | 32.1 |
| | (31.1–50.9) | (28.3–47.2) | | | | | | |
| Tail L | 108.4 ± 8.6 | 114.2 ± 6.3 | 34.8 ± 3.1 | 34.2 ± 2.9 | 35.8 | 113.2 | 33.0 | 109.4 |
| | (91.5–119.8) | (103.7–125.4) | (30.2–43.4) | (30.2–39.6) | | | | |
| St L/LRW | 1.5 ± 0.2 | 1.42 ± 0.1 | 1.4 ± 0.2 | 1.5 ± 0.1 | 1.6 | 1.5 | 1.5 | 1.5 |
| | (1.2–2.0) | (1.1–1.6) | (1.1–1.6) | (1.4–1.8) | | | | |
| Corpus L / isthmus L | 1.77 ± 0.2 | 2.5 ± 0.3 | 1.7± 0.3 | 2.5± 0.2 | 1.6 | 1.6 | 2.4 | 2.6 |
| | (1.5–2.4) | (2.0–3.2) | (1.5–2.8) | (2.2–2.8) | | | | |
| Nerve ring (% PL) | 68.3 ± 0.03 | 72.7 ± 0.03 | 67.2 ± 0.03 | 73.1 ± 0.02 | 69.0 | 64.1 | 70.5 | 73.0 |
| | (61.1–72.6) | (66.8–79.7) | (61.9–74.6) | (70.5–75.9) | | | | |
| E-pore (% PL) | 90.6 ± 0.03 | 92.1 ± 0.02 | 95.1 ± 0.05 | 90.6 ± 0.04 | 92.0 | 69.4 | 95.8 | 94.0 |
| | (87.8–97.1) | (86.7–96.7) | (85.0–100.0) | (84.4–96.3) | | | | |
| G (no flex as % L) | 57.6 ± 4.2 | 45.4 ± 0.7 | – | – | – | 58.3 | – | 45.1 |
| | (52.4–65.3) | (25.1–57.0) | | | | | | |
| Rectum L / ABD | 1.0 ± 0.1 | 1.03 ± 0.2 | – | – | – | 1.0 | – | 1.1 |
| | (0.8–1.3) | (0.8–1.4) | | | | | | |
| Phasmid (% TL) | 37.2 ± 6.7 | 24.5 ± 0.04 | – | – | – | 34.2 | – | 29.3 |
| | (30.3–50.5) | (24.2–39.4) | | | | | | |
| Spicule L | – | – | 60.4 ± 5.1 | 58.5 ± 3.0 | 48.1 | – | 62.2 | – |
| | | | (48.1–67.9) | (53.8–62.2) | | | | |
| Gubernaculum L | – | – | 27.7 ± 2.9 | 19.0 ± 2.4 | 21.7 | – | 19.8 | – |

(*Continued*)

**Table 1.** (*Continued*)

| Character | *P. pelhamensis* n. sp. EM434 ♀[1,2] | *P. pellio* SB361 ♀ | *P. pelhamensis* n. sp. EM434 ♂[1,2] | *P. pellio* SB361 ♂ | *P. pelhamensis* n. sp. Holotype ♂ | *P. pelhamensis* n. sp. Paratype♀ | *P. pellio* Neotype ♂ | *P. pellio* Topotype ♀ |
|---|---|---|---|---|---|---|---|---|
| | | | (21.7–31.1) | (15.1–23.6) | | | | |

[1] Grown on nematode growth medium with *Ochrobactrum* sp.

[2] Measurements were made from 4% formalin-fixed specimens processed to anhydrous glycerin, following a modified procedure of Seinhorst [25] as modified by De Grisse [24]. $G_1/G_2$: Vulva to anterior/posterior flexure of gonad as % of body length in female; corpus length (CL): measured along curvature of the lumen; neck length (NL): from anterior end to the base of the basal bulb, measured along middle of the body; reproductive tract length (RTL): measured along body axis, from anterior-most tip to posterior-most tip, *i.e.*, excluding all flexures; stoma length: from cheilorhabdia to base of the stoma. Stoma terminology (cheilostom, gymnostom, and stegostom) was adapted from De Ley *et al.* [60], and terminology associated with the structures of the nematode anterior is based on Rashid *et al.* [61].

[3] After Hooper *et al.* [62].

[4] Equivalent to pharynx measurement in Hooper *et al.* [62]

type location for this species [18]. The strain was cultured on water agar plates supplemented with small pieces of beef; first cryogenically preserved June 17, 2005, by K. Kiontke.

CH1: leg. J. Nermuť in summer, 2012, from a gray slug, *Deroceras reticulatum* Müller, 1774, from a redcurrant plantation near the village of Chelčice, Czech Republic (49.1287125, 14.1739622), altitude 450 m above sea level [6]. SSU rDNA sequences of haplotype 1 of CH1 and SB361 are identical (haplotype 2 is different from haplotype 1 by 11 nucleotides in a cluster); LSU rDNA and ITS sequences of CH1 and SB361 are identical except for 2 ambiguities and 1 SNP (single nucleotide polymorphism).

185: leg. J. Nermuť in summer, 2012, from a gray slug, *Deroceras reticulatum* Müller, 1774, from a redcurrant plantation near the village of Chelčice, Czech Republic, close to where CH1 was isolated [6]. LSU rDNA and ITS sequences of CH1 haplotype 1, 185 and SB361 are identical except for 2 ambiguities and 1 SNP.

**Redescription.** Based on strain SB361.

*Illustrations*. Figs 5–8, supplemental videos showing stacks of DIC focal planes through live animals (S10–S19 Videos in S2 File).

*Measurements*. See Table 1.

*Adult females and males*. Females and males are very similar to *P. pelhamensis* n. sp., but about 5% longer and 40% wider at maximum body width when grown under the same conditions. A clear difference is found in the stoma, which is consistently longer and narrower in *P. pellio*: the stoma of *P. pellio* is about 4.5 times as long as wide (Figs 5D, 6E, 8E and 8F), whereas that of *P. pelhamensis* n. sp. is less than 3 times as long as wide (Figs 1D, 2B, 8A and 8B). A subtle difference is seen in the tip of the spicule, which is narrower in *P. pelhamensis* n. sp. than in *P. pellio* (Figs 8C, 8D, 8G and 8H). In strain SB361, there is variation in ray positions such that GP v4 and the anterior dorsal GP may form a close group (Figs 5F, 6I and 7L–7N) or not (Fig 6G and 6H). Aberrations with fused or missing GPs are frequently observed. All other dimensions and morphological features are essentially identical in both species.

*Life cycle*. Here, we synthesize several observations of the *P. pellio* life cycle from different sources. *P. pellio* is a necromenic bacteriophagic nematode, whose dauer juveniles invade nephridia, seminal vesicles or the coelom of different earthworm species or become encapsulated in brown bodies [9, 63]. There, they wait for the earthworm to die and develop on the cadaver or develop in the decaying terminal end of the worm when it is detached by autotomy [64]. Dauer juveniles do not wave and do not survive dessication. However, they have survived in water 4½ months at room temperature [65] and 24 months at 2.5 ˚C (WS, unpub. observations). Males are nearly as frequent as females (43–48%). Copulation follows the parallel type

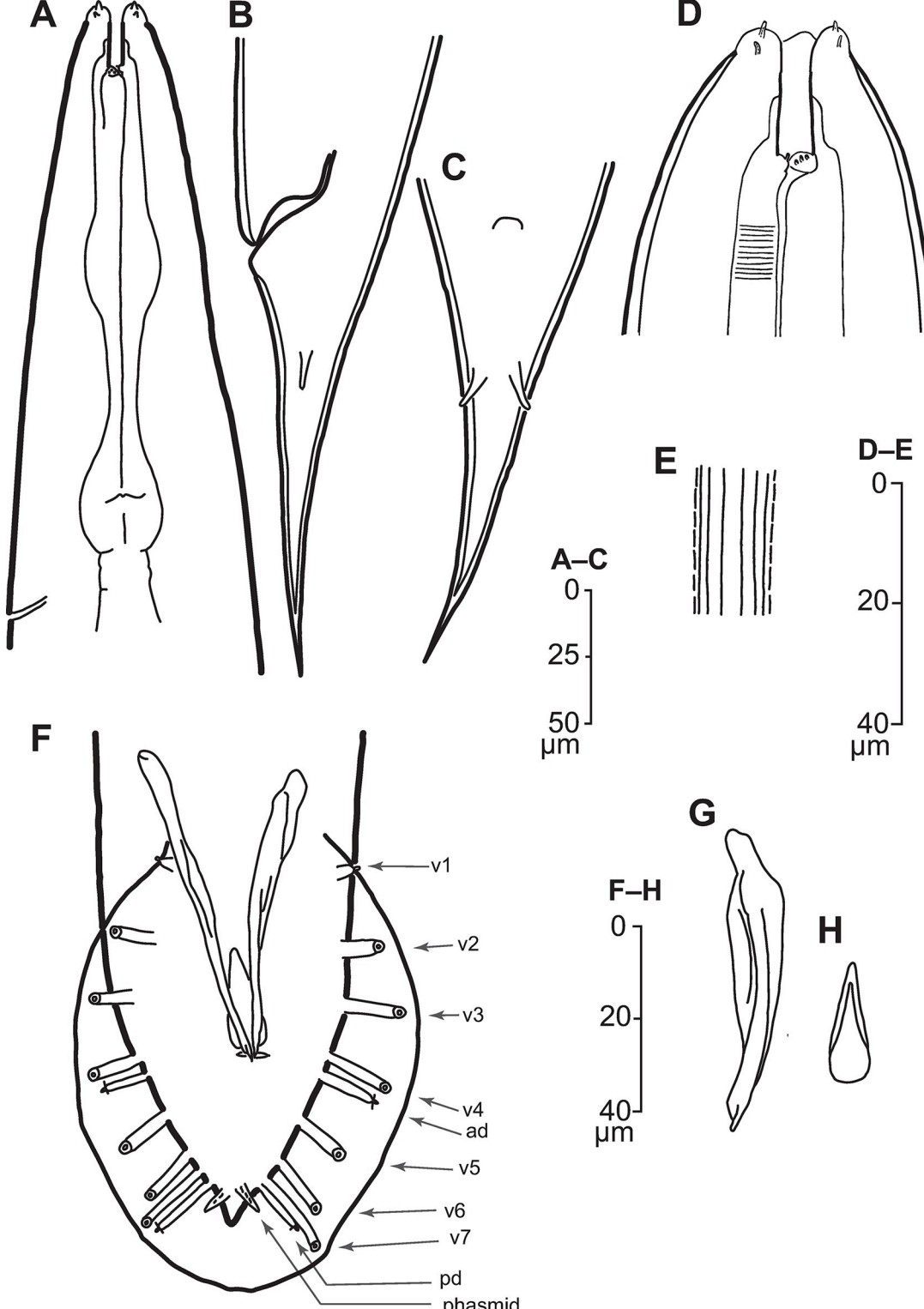

**Fig 5.** Drawings of *Pellioditis pellio* strain SB361 female (A-E) and male (F-H). (A) Anterior portion showing stoma, pharynx and excretory pore, left side view. (B, C) Female tail showing positions of anus and phasmids; left side (B) and ventral (C) views. (D) Stoma region, right side view. (E) Lateral field near the midbody. (F) Ventral view of adult male tail with genital papillae labeled according to the rhabditid homology system [28, 29] based on common developmental origins of the GPs in the lateral epidermis [23, 27, 58, 59]. (G) Spicule (lateral view) and (H) gubernaculum (ventral view). Scale bars as shown.

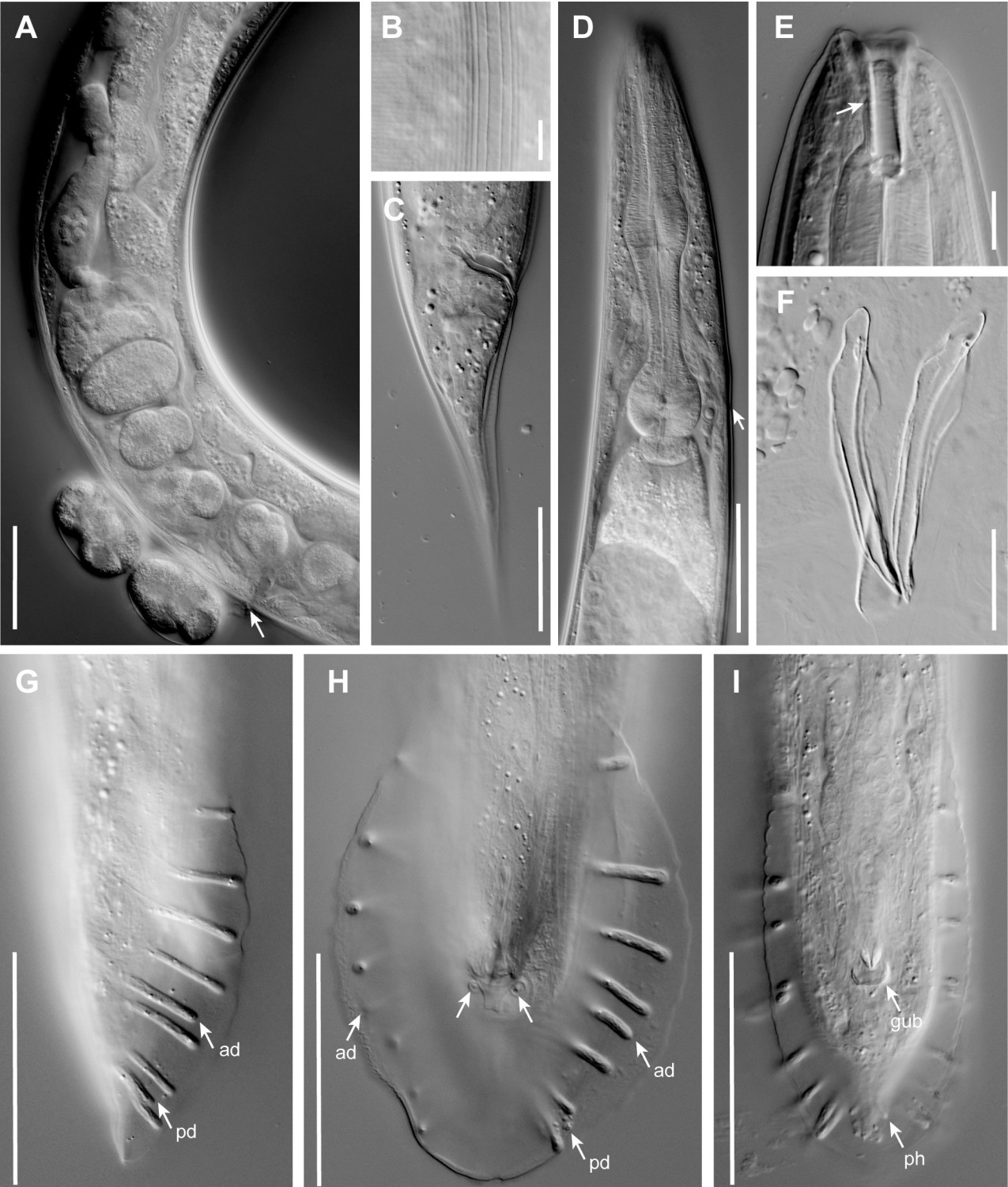

**Fig 6. DIC photomicrographs of *P. pellio*.** (A) Anterior uterus and oviduct of a young female and two laid embryos, arrow points to the position of the vulva; left side view, anterior up. (B) Female lateral field. (C) Female tail in right side view. (D) Pharynx region of an adult male, right side view; arrow points to the location of the excretory pore. (E) Stoma of an adult female in right side view; arrow points to the border between gymnostom and stegostom. (F) Isolated spicules in lateral view and gubernaculum in ventral view. (G) Male tail in right side view. Anterior (ad) and posterior dorsal (pd) genital papillae are indicated with arrows. (H) Male tail in ventral view; the base of the postcloacal sensilla is in focus here (unlabeled arrows). (I) Male tail in ventral view, focused closer to the body than in H and showing phasmids (ph) and the distal end of the gubernaculum (gub). Scale bars 10 μm in B and E, all others 50 μm.

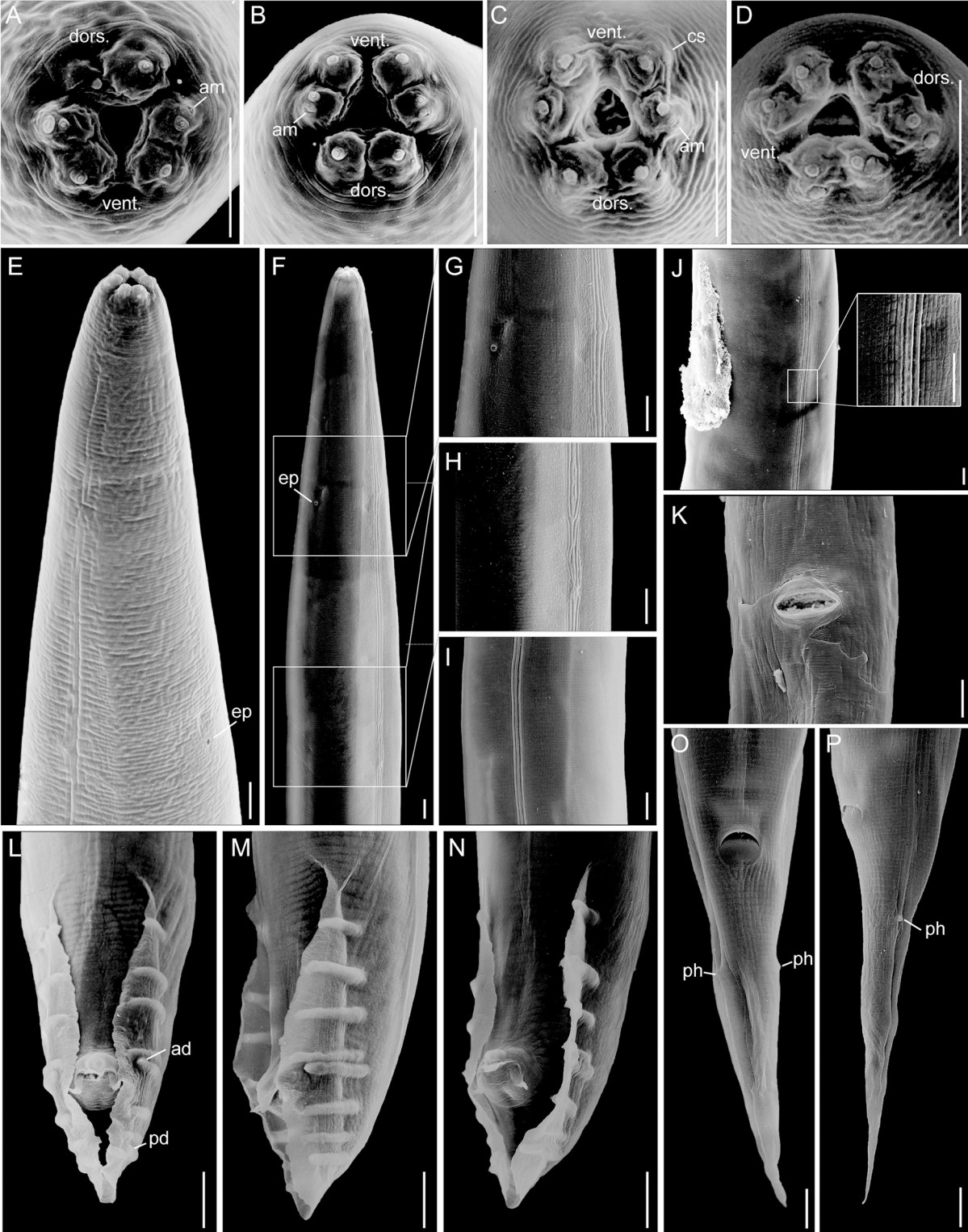

**Fig 7.** SEM micrographs of *Pellioditis pellio* females (A, B, E, J, K, O, P) and males (C, D, F-I, L-N). (A, B) Lip region of female, *en face* view; in A, the right dorsal lip is abnormal. (C, D) Lip region of male, *en face* view. (E) Anterior portion of a female in subventral right view. (F) Anterior portion of a male in subventral left view, (G–H) enlarged detail views corresponding to areas delineated in F. (I) Lateral field at mid-body in male. (J) Vulval region of a female with mating plug; inset shows detail of lateral field. (K) Female vulva. (L–N) Male tail in ventral or subventral view showing narrow bursa with genital papillae and cloacal opening with precloacal sensillum and postcloacal sensilla. (O, P)

Female tail in ventral and left side views. The phasmids (ph) form small papillae. Abbreviations: am = amphid opening, cs = cephalic sensilla, ep = excretory pore, ad = anterior dorsal genital papilla, pd = posterior dorsal genital papilla, ph = phasmid. Scale bars 10 μM.

and a mating plug is deposited (Fig 7J). The species is ovoviviparous or juveniles develop inside the mother up to the dauer stage. Development takes 3.5–4 days at 20 ˚C, generation time is 4–5 days, life span of adults 5–10 days. Each female can have hundreds of offspring; e.g. 570 offspring were reported from one female mated several times [66].

*Identification.* Strain SB361 is no different from the original description of *P. pellio* [18] (cf. Fig 9). Like the original isolate, it was obtained from a gray earthworm from the type location (Berlin, Germany); morphological features are identical: e.g. a conical female tail with a thick cuticle and prominent phasmids, male ray and bursa morphologies. *P. pellio* is reported to be commonly isolated from earthworms [9, 18, 67–69] (Fig 9) and is likely necromenic on them [11]. Thus, strain SB361—isolated from the type location and host—represents *P. pellio*.

*Voucher material.* Of strain SB361, a slide with the neotype male (morphometrics in Table 1) has been deposited at Wageningen Nematode Collection as slide WT3937. This slide has three males and three females from the series; the neotype is the leftmost male. A slide with the measured topotype female (Table 1) has been deposited with Ghent University Zoology

## *P. pelhamensis* n. sp.    *P. pellio*

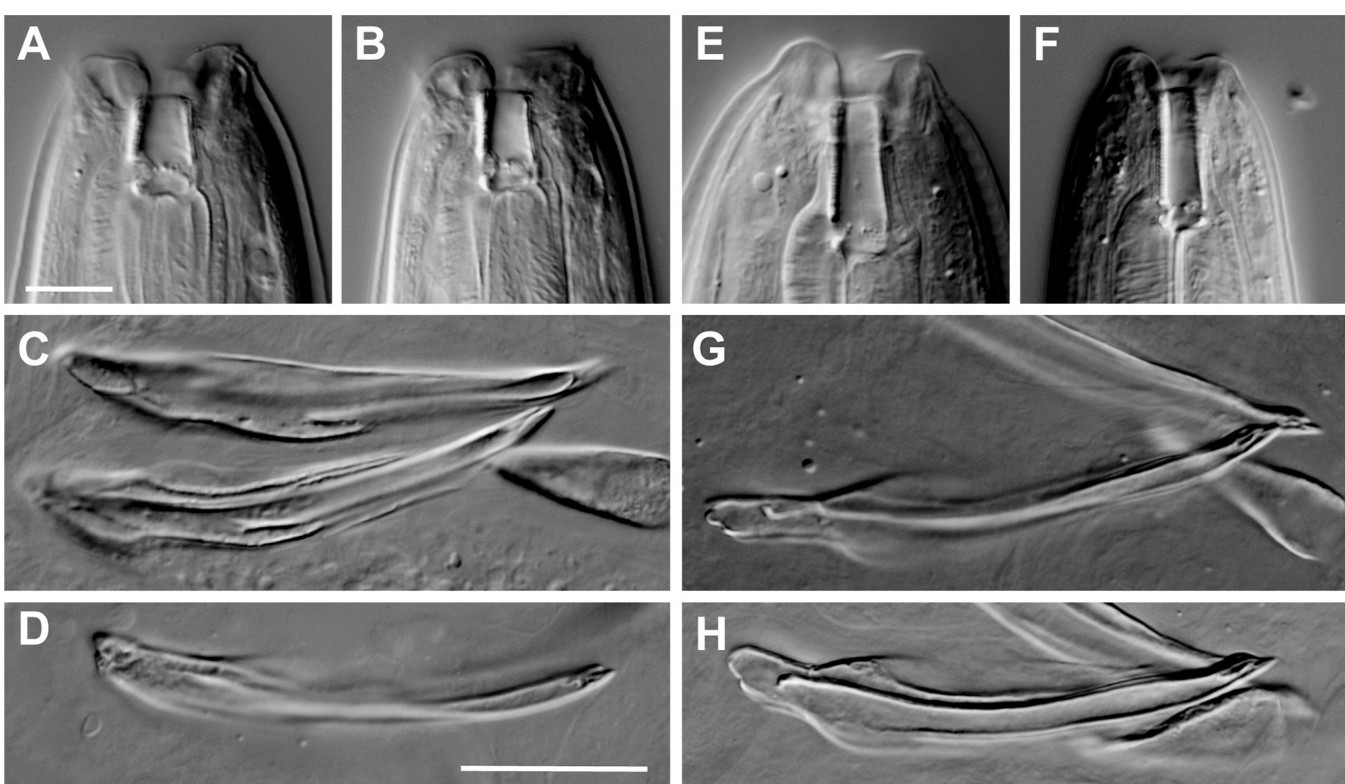

**Fig 8. DIC photomicrographs comparing buccal tube and spicules of *Pellioditis pelhamensis* n. sp. and *Pellioditis pellio*.** *P. pelhamensis* n. sp. and *P. pellio* differ in the shape of the stoma and in the shape of the spicule tip. The stoma of *P. pelhamensis* n. sp. is shorter and wider—and the spicule tip is narrower—than those of *P. pellio*. (A–D) *P. pelhamensis* n. sp. (E–H) *P. pellio*. (A) Stoma of an adult female in subventral left view. (B) Stoma of an adult male in subventral left view. (E) Stoma of an adult female right side view. (F) Stoma of an adult male right side view. (C, D, G, H) Spicules in lateral view; gubernaculum in ventral view. Scale bars for all stoma images 10 μm (A, B, E, F), for all spicule images 20 μm (C, D, G, H).

## Original *P. pellio* and plausible reisolates

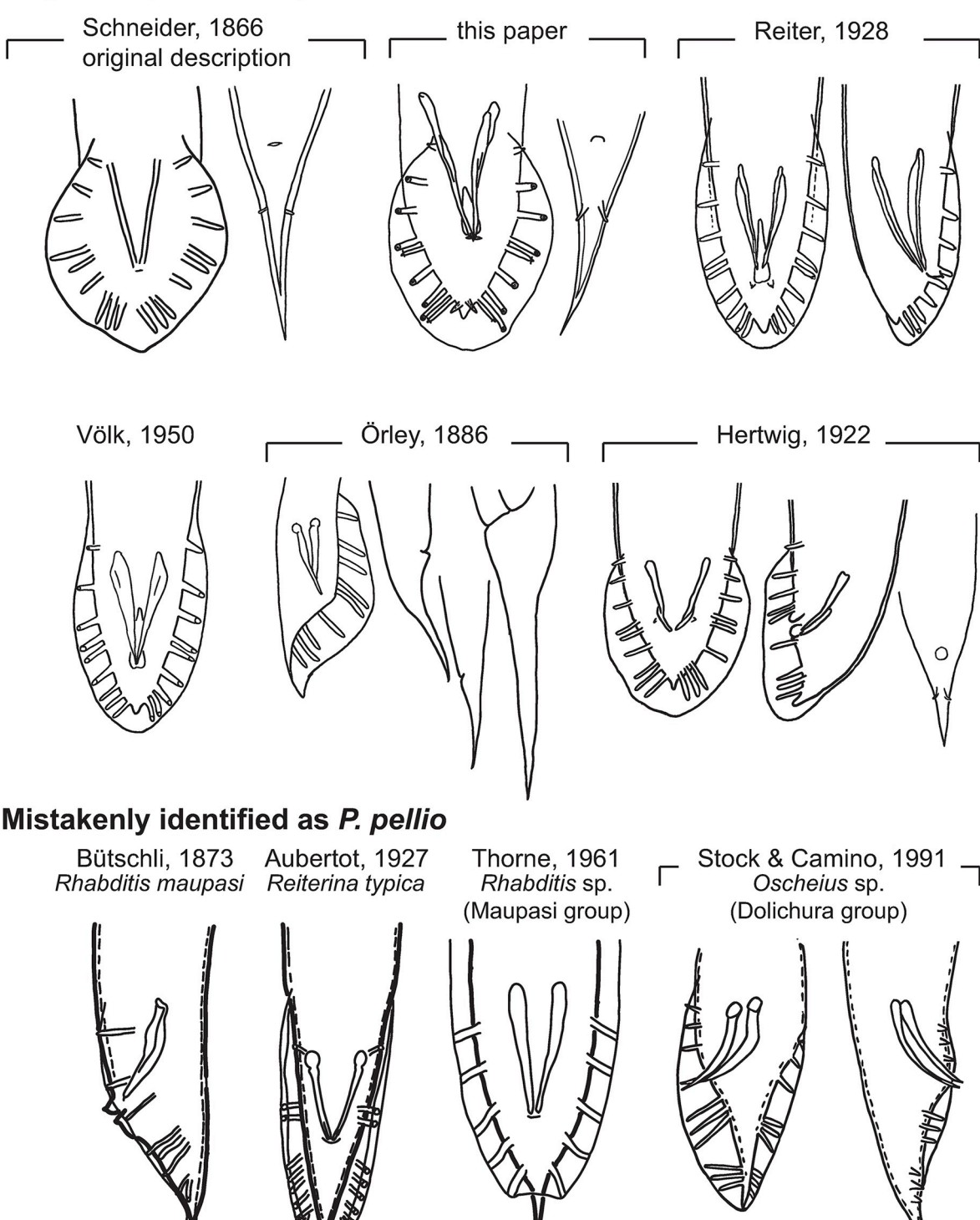

## Mistakenly identified as *P. pellio*

**Fig 9. Comparison of male and female tails of *Pellioditis pellio* redescribed here with those depicted in figures from the original description (Schneider, 1866) [18], subsequent plausible reisolates and isolates mistakenly claimed to be *P. pellio*.** Schneider [18] only provided a single drawing in the original description: a female tail in ventral view. In the description of *P. pellio*, he noted that the species was in all aspects similar to his directly preceding description of *P. papillosa* (of which the male bursa is redrawn here from Taf. XI, Fig IIIc [18]), except that the female tail of *P. papillosa* was cupola-shaped whereas that of *P. pellio* was cone-shaped (redrawn here from Taf. XI,

Fig X [18]). Neither Schneider nor any of the other authors depicted the phasmids in the males, which are admittedly not very prominent in this species and could have easily been overlooked. In all of the depictions of species compatible with the original description (e.g., from the current paper, Reiter [68], Völk [69], Örley [67], and Hertwig [9]), the male tail is peloderan and GPs 4 and 5 (i.e., v4 and ad) are situated close together, at about the same anteroposterior location as the cloaca. Also, wherever female tails are depicted in these descriptions, all show conical forms, though of varying lengths consistent with variation present in strain SB361 (Table 1). In contrast, the male tails from other depictions mistakenly identified as *P. pellio* (bottom row) have clearly different male tail tip morphologies and papilla arrangements. For example, the drawings by Bütschli [70] and Thorne [71] show species with leptoderan tail tips; those by Bütschli [70] and Aubertot [72] show GPs 2–3 bracketing the cloaca and those by Thorne [71] and Stock & Camino [73] show GPs 3–4 bracketing the cloaca.

Museum, as slide UGMD_104443. This slide has three females and three males from the series; the measured topotype female is leftmost. A third slide with 3 males and four females of this series has been deposited as slide UCDNC 5259 at the Nematode Collection, University of California, Davis. Live cultures of SB361 are cryogenically archived at the CGC and the NYURC.

*DNA sequences deposited at GenBank for SB361*. OR059187 (newly reported here): nearly complete repeat unit for the nuclear ribosomal RNA genes (18S, ITS1, 5.8S, ITS2, 28S).

## Differential diagnosis

*P. pelhamensis* n. sp. and *P. pellio* can be distinguished from each other and from all known species of the *Pellioditis* clade. As noted above, females and males of *P. pellio* are slightly larger than *P. pelhamensis* n. sp. when grown under the same conditions. The stoma is shorter and wider in *P. pelhamensis* n. sp. than in *P. pellio* (Fig 8A, 8B vs. 8E and 8F) and the spicule tip is slightly narrower in *P. pelhamensis* n. sp. than in *P. pellio* (Fig 8C, 8D vs. 8G and 8H). We also performed reciprocal crosses (Table 2). These crosses demonstrated that (1) *Pellioditis pelhamensis* EM434 is reproductively isolated from *Pellioditis pellio* SB361, and (2) isolation is postzygotic, as sperm from each species were able to induce egg maturation (e.g. production of eggshell). We did not check for stages at which embryonic development was arrested, however.

Two species of *Pellioditis* have only rare or no males: *P. hermaphrodita* and *P. californica* [54], whereas approximately half of *P. pelhamensis* n. sp. and *P. pellio* populations are males (although a parthenogenetic variant of *P. pellio* has been documented [9]).

In contrast to the conical-shaped female tails of *P. pelhamensis* n. sp. and *P. pellio*, many *Pellioditis* species have female tails that are moderately or strongly cupola-shaped (dome-shaped with a pointed spike projecting from the "top" of the dome): *Pellioditis akhaldaba*, *Pellioditis bonaquensis*, *Pellioditis huizhouensis*, *Pellioditis incilaria*, *Pellioditis meridionalis*, *P. papillosa*, *Pellioditis safricana*, and *P. zhejiangensis*. From its description [74], it appears that *P. akhaldaba* actually encompasses two different but closely related species (see the list of valid species below). One form ("slow") has mostly cupola-shaped tails, whereas the other form ("quick") has conoid-tails. This latter form can be distinguished from *P. pelhamensis* n. sp. and *P. pellio* by having, for example, different DNA sequences and a lateral field with three strong ridges (instead of ≥5).

**Table 2. Interfertility tests[1].**

|  | No males | EM434 males | SB361 males |
|---|---|---|---|
| EM434 females | many oocytes laid without eggshells | many juveniles (and some young adults) | eggs only (with eggshells) |
| SB361 females | no oocytes detected | eggs with eggshells and some oocytes | many juveniles (and some young adults) |

[1] See Materials and Methods for details of how crosses were set.

*P. pellio* and *P. pelhamensis* n. sp. can also be differentiated from other gonochoristic species having conical or conoid female tails, i.e. *P. apuliae*, *P. bohemica*, *Pellioditis circassica*, *Pellioditis clausiliiae*, *Pellioditis kenyensis*, *P. neopapillosa*, *Pellioditis quinamensis*, *Pellioditis thesamica*, *Pellioditis tawfiki*, and *Pellioditis villasmundi*. *P. tawfiki* males have GP1 and GP2 very close together, and only these GPs are precloacal [75]; Sudhaus [76] has previously noted this species fits better in *Pelodera* than in *Pellioditis*. In males of *P. kenyensis* [48], GP4 and GP5 are the same distance apart as GP3 and GP4, different from the usually close apposition between GP4 and GP5 in *P. pelhamensis* n. sp. and *P. pellio*. In *P. thesamica* [56], GP1 and GP2 are considerably further apart than are GP2 and GP3, whereas these distances are similar in most individuals of *P. pelhamensis* n. sp. and *P. pellio*. In *P. quinamensis* [47], each of the three pairs of lips are strongly fused and separated from the other pairs, different from our strains where there is distinct separation between lips of each pair. Although *P. villasmundi* is morphologically similar to *P. pelhamensis* in many ways (including a short buccal tube), there is greater space between GP1 and GP2 than between GP2 and GP3 (GP1–3 are similarly spaced in *P. pelhamensis* and *P. pellio*), and they differ substantially at the molecular level (26, 24 and 3.3% different at ITS1, ITS2 and LSU, respectively). Four species, *P. apuliae*, *P. bohemica*, *P. circassica*, and *P. clausiliiae* [6, 16, 46], all have a similar GP formula and can only be distinguished by DNA sequences and morphometrics. Two of these, *P. circassica* and *P. clausiliiae* have the shortest female tails and the shortest male spicules (<41 μm) [46]; all the others, *P. apuliae*, *P. bohemica*, *P. pelhamensis* n. sp. and *P. pellio*, have longer female tails on average and much longer male spicules (>48 μm). *P. apuliae* [16] and *P. pelhamensis* n. sp. have short, wide stomas (length/width ratios <3), whereas *P. bohemica*, *P. pellio* and the others have longer stomas (length/width ratios >3).

Thus, the only two species not differentiable on morphological grounds are *P. bohemica* and *P. pellio*. As detailed below, they are also not differentiable at the molecular level and are therefore synonymous. That is, *P. bohemica* is a junior synonym of *P. pellio*. The fact that *P. bohemica* was collected from slugs accords with previous accounts of *P. pellio* being isolated from slugs as well as earthworms [5, 10, 65].

One further species to compare, *Pellioditis mairei* [5], is poorly described, with only a figure of the male tail and measurements for one female. Drawings of the male bursa look essentially identical to those of *P. pellio*. Two characters that distinguish *P. mairei* from *P. pellio* (and strain SB361) as well as *P. pelhamensis* are the much larger size of *P. mairei* females (3400 μm vs. ca. 1500 μm) and the relatively shorter (presumably conoid) female tail (c = 23.6 for *P. mairei* vs. ca. 13–14). Strain DD of *P. mairei* was also isolated from an earthworm, but from a garden in Algiers [5]. Reciprocal crossing experiments showed that *P. mairei* was postzygotically isolated from *P. pellio* collected from slugs [2, 5].

## Notes on erroneous and plausible records of *P. pellio* in the literature

Because we re-isolated *P. pellio* from its type location and could study its characters carefully, we can scrutinize previous records of *P. pellio* (Fig 9). For early nematologists, a preferred method for obtaining rhabditid nematodes was to chop earthworms into heat-treated soil and wait for necromenic nematodes to develop on the putrefying cadaver. Often, these authors assumed that the nematodes they isolated were *P. pellio* without checking morphological characters, even though several species of rhabditids are associated with earthworms and *P. pellio* is not even the most frequent [77]. Fortunately, some authors provided figures or references to characters, thus allowing subsequent identification of the actual species under investigation. For example, many authors who referred to a species called *"Rhabditis pellio"* as described by Bütschli, 1873 [70] actually must have been working with *Rhabditis maupasi* Seurat in Maupas,

1919 [5]. This species is clearly distinguished from *P. pellio* by a leptoderan versus peloderan male tail tip (Fig 9). The species investigated by Aubertot [72] under the name *"R. pellio"* can be identified via his clear illustration as *Reiterina typica* Stefański, 1922 (Fig 9). Consequently, all notes about *"R. pellio"* in the literature referring to Aubertot's observations that *"R. pellio"* has waving dauer juveniles and phoresis on *Drosophila* should instead be applied to *R. typica* and not *P. pellio*. Another example is the "*Rhabditis (Pelloides) pellio*" (*Pelloides lapsus*) described by Thorne, 1961 (p. 451 in [71]) (Fig 9). This specimen is clearly not *P. pellio* as originally described by Schneider, since it has a leptoderan male tail tip. Instead, it belongs to the *Maupasi* species group of *Rhabditis*. Because Thorne obtained his specimen from Dougherty [71], we also have to be somewhat skeptical that Dougherty & Calhoun [78] were working with *P. pellio*, even though Chitwood identified their culture as *"R. pellio"* (see footnote 2, p. 55, of Dougherty & Calhoun [78]). As a final example of misidentification, the alleged "*Pellioditis pellio*" isolated from tabanid larvae by Stock & Camino [73, 79] is in all likelihood an *Oscheius* species (Fig 9).

On the other hand, several other investigators could have plausibly worked with the "true" *P. pellio*, or at least a very similar *Pellioditis* species. The illustrations associated with the studies of Reiter, Völk, Örley, and Hertwig [9, 67–69] all suggest they could well have been working with *P. pellio* (Fig 9). Even without illustrations, it is sometimes clear that *P. pellio* or a very closely related species was used. Poinar & Thomas [80] and Somers et al. [66] investigated a rhabditid they called *"Rhabditis pellio"* from the earthworm *Aporrectodea trapezoides* in California. From their introduction, it is clear that these authors were aware of the differences between Bütschli's leptoderan "*R. pellio*" and the actual peloderan *P. pellio* of Schneider [80]. Poinar identified the species that Eveland et al. [81] used in their scanning electron microscopy investigation, which was thus likely *P. pellio* or a similar species.

Finally, because they recorded key morphological characters, we can now determine that other investigators claiming to use *P. pellio* actually used instead a closely related *Pellioditis* species. According to the male bursa and other characters they described, Goodchild & Irwin [82] identified an isolate very similar to *P. pellio* from *Lumbricus* in Georgia and Illinois. However, their clear picture of the stoma shows a length/width ratio <3, more consistent with *P. pelhamensis* n. sp. than with *P. pellio* (length/width ratio 4.5). Similarly, Zaborski et al. [83] grew a nematode species from *Lumbricus terrestris* from Illinois that is similar to *P. pellio* in some respects, but has a short and wide stoma and is as large as *P. mairei* (female length >3 mm) [5].

## Valid species in genus *Pellioditis* Dougherty, 1953

*Pellioditis* Dougherty, 1953: type species *Pelodera pellio* Schneider, 1866 [18, 84].

syn. *Phasmarhabditis* Andrássy, 1976: type species *Pelodera papillosa* Schneider, 1866 [11, 85].

Sudhaus [2, 11] previously combined *Pellioditis* instead of *Phasmarhabditis* with species names below.

- *P. akhaldaba* (Ivanova, Gorgadze, Lortkhipanidze & Spiridonov, 2021)—Akhaldaba, Georgia [74]. Note: It is likely that this species name was applied to two very similar but different species from the same locality, one from the slug *Deroceras reticulatum* and one from garden soil baited with *Galleria* (wax moth) larvae. The labeling of the different isolates as "quick" (from slugs) and "slow" (from soil) reflects a behavioral difference. The "quick" females have a conical tail whereas the tail is cupola-shaped in 80% of the "slow" females. The "quick" isolate is generally smaller than the "slow" one, and several measurements show significant differences between the isolates: stoma length, diameter of corpus and terminal bulb of the

pharynx, flexure of the testis, distance between anus and phasmids in females and distance between lips and excretory pore in exsheathed dauer larvae. Unfortunately, no crossing experiments were reported to test interisolate reproductive compatibility.

- *P. apuliae* (Nermuť, Půža & Mráček, 2016)—Bari, southern Italy; from the slug *Milax sowerbyi* in the garden of the University [16].

- *P. bonaquensis* (Nermuť, Půža, Mekete & Mráček, 2016), emendation of *Phasmarhabditis bonaquaense* Nermuť, Půža, Mekete & Mráček, 2016 by D. J. Hunt (footnote on p. 232 of [47])—České Švýcary, near Dobrá Voda, Czech Republic; from the slug *Malacolimax tenellus* [15].

- *P. californica* (Tandingan De Ley, Holovachov, Mc Donnell, Bert, Paine & De Ley, 2016)—Eureka, California, USA; obtained from a cadaver of the slug *Deroceras reticulatum* [54].

- *P. circassica* (Ivanova, Geraskina & Spiridonov, 2020)—Nickel settlement, Adygea Republic, Russia; from the snail *Oxychilus* cf. *difficilis*, in deciduous forest [46].

- *P. clausiliiae* (Ivanova, Geraskina & Spiridonov, 2020)—near Georgievsk, Stavropol district, Russia; from Clausiliidae snails (cf. *Quadriplicata* sp.) in deciduous forest [46].

- *P. hermaphrodita* (Schneider, 1859)—Berlin, Germany; in a slug (or snail? "Schnecke") [86].

  syn. *Leptodera foecunda* Schneider, 1866 [18]
  *Rhabditis caussaneli* Maupas, 1899—Vire, Normandy, France; dauer juveniles in the slug *Arion empiricorum* var. *ater* [65].

- *P. huizhouensis* (Huang, Ye, Ren & Zhao, 2015)—Kowloon Peak of Huizhou City, Guangdong Province, China; in rotting leaves [13].

- *P. incilaria* (Yokoo & Shinohara in Shinohara & Yokoo, 1958)—Kurume-Shi, Kyushu, Japan; isolated from the slug *Incilaria confusa* [87].

  syn. *Rhabditis fruticicolae* Kreis, 1967, *nec* Shinohara, 1960 [88]—Nagasaki, Japan; from the snail *Fruticicola sieboldiana* [89].

- *P. kenyensis* (Pieterse, Rowson, Tiedt, Malan, Haukeland & Ross, 2021)—Nairobi, Kenya; from the slug *Polytoxon robustum* in a garden [48]. Note: *P. kenyensis* is a new emendation of *P. kenyaensis*, as per ICZN Article 32(d) and exemplified in Appendix D, sec. IV, item 22 (a).

- *P. mairei* (Maupas, 1919)—locality not given, but likely in Algeria; dauer juveniles in a gray earthworm from a garden [5].

- *P. meridionalis* (Ivanova & Spiridonov, 2017)—Cát Tiên National Park, southern Vietnam; from the land snail *Quantula striata* [90].

- *P. neopapillosa* (Mengert in Osche, 1952)—Bavaria (near Berchtesgaden or near Hersbruck), Germany; dauer juveniles in the keelback slug *Limax cinereoniger* [91].

- *P. papillosa* (Schneider, 1866)—Berlin, Germany; in decaying substance, dauer juveniles associated with the slug *Limax ater*; necromenic in slugs, Arionidae and Limacidae [18].

  syn. *Angiostoma limacis apud* Will (1848) [92], *nec* Dujardin, 1844 [93].
  *Rhabditis ikedai* Tadano, 1950—Sendai or Tokyo, Japan; from the slug *Incilaria confusa* [94].

*Rhabditis ninomiyai* Yokoo, 1968—Nagasaki, Japan; from the alimentary organs of a snail (*Fruticicola* or *Euhadra*) [95].

- *P. pelhamensis* Tandingan De Ley, Kiontke, Bert, Sudhaus & Fitch, 2023 n. sp.—Bronx, New York and Oregon, USA; from earthworms and slugs (*D. reticulatum*, *Testacella* sp.).

- *P. pellio* (Schneider, 1866)—Berlin, Germany; dauer juveniles in the body cavity of a gray earthworm [18]. Due to carelessness, very often confused with other species from earthworms, especially *Rhabditis maupasi*. (See text above for prior mistaken identifications of *P. pellio*.)

   syn. nov. *Pellioditis bohemica* (Nermuť, Půža, Mekete & Mráček, 2017)—near Chelčice, Czech Republic; from *Deroceras reticulatum* from a red currant plantation; also from soil samples baited with *Galleria* larvae [6].

- *P. quinamensis* (Ivanova & Spiridonov, 2022)—Cát Tiên National Park, southern Vietnam; from the land snail *Sesara* sp. Further isolates from seven additional genus taxa of terrestrial snails in that National Park [47].

- *P. safricana* (Ross, Pieterse, Malan & Ivanova, 2018)—near George, Western Cape province, South Africa; from the slug *Deroceras reticulatum* from a nursery [53].

- *P. thesamica* (Gorgadze, Troccoli, Fanelli, Tarasco & De Luca, 2022)—Tezami, East Georgia; from a dead *Deroceras reticulatum* slug in a raspberry garden [56].

- *P. villasmundi* (Ivanova, Clausi, Leone & Spiridonov. 2023)—Nature Reserve 'Speleological Complex Villasmundo—S. Alfio', Syracuse Province, Sicily; from the slug *Milax nigricans*, found in 7 additional gastropod species [96].

- *P. zhejiangensis* (Zhang & Liu, 2020)—Ningbo, Zhejiang province, China; from the slug *Meghimatium (= Philomycus) bilineatum* from a vegetable garden [97].

   Incertae sedis:

- *P. tawfiki* (Azzam, 2003)—Great Cairo, Egypt; unclear whether the type was isolated from the snail *Eobania vermiculata* or the slug *Limax flavus* [75]. Note: due to several described characters, Sudhaus [2] suggests that this dubious species may instead belong in genus *Pelodera* (*Teres* species group).

## Phylogenetic relationships using DNA sequences

Most *Pellioditis* species (19 described and 5 undescribed) are represented by DNA sequences covering various segments of nuclear and mitochondrial ribosomal RNA genes, part of the gene for the large subunit of RNA polymerase II (minus introns), and/or the mitochondrial cytochrome oxidase I gene and other various mitochondrial genes. The phylogenetic diagrams that have been published with species descriptions since 2015 are usually only based on small sequence segments from individual genes and vary widely in the relationships depicted. While these analyses may serve to show genus-level membership of a new species, building a robust phylogenetic framework for testing evolutionary hypotheses requires much more extensive molecular data.

   Here, we collected as many representative sequences as possible from GenBank (200; S1 Table) and constructed a DNA alignment supermatrix (see Materials and Methods for details). To this matrix, we added our two new nearly complete rDNA sequences for *P. pelhamensis* n. sp. EM434 and *P. pellio* SB361. We included an *Agfa* and an *Angiostoma* species, proposed to

be closely related to *Pellioditis* [11, 28, 98], as well as representatives from nearly every genus in Rhabditidae for an outgroup sample. To test how robust the phylogenetic inference would be to methodology, we analyzed the data via two different approaches: (1) weighted parsimony (WP) using the full dataset (all taxa with all characters), which contained a lot of gaps or missing data, and (2) maximum likelihood (ML) using taxon- and character-subsets of the supermatrix, each with few gaps, and then puzzling the resulting trees into a supertree. Bootstrapping (for ML) and jackknifing (for WP) were used to test how robust inferred bipartitions (phylogenetic branches) are to data sampling.

Although the ML approach provided less resolution than WP, phylogenies using both approaches are compatible in most respects (Fig 10A). Conflict is only apparent in how the three *Pellioditis* species groups are related (see below) and in two of the deeper relationships within the outgroup. These cladograms allow us to make the following observations:

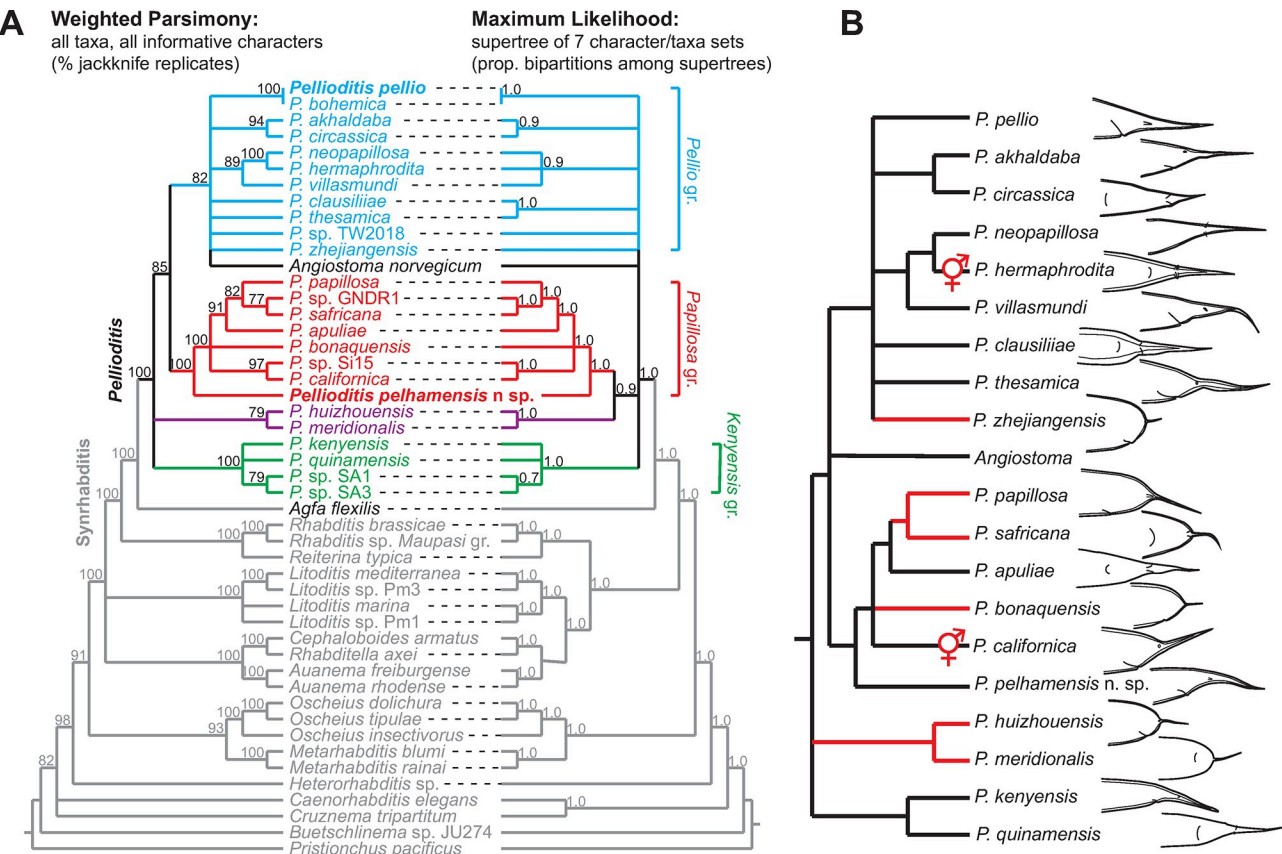

**Fig 10.** Results of phylogenetic analyses (A) and evolutionary reconstructions (B). (A) Molecular phylogenetic analyses. Left tree: WP analysis of all parsimony-informative sites in the supermatrix, which consists of sequences from the nuclear ribosomal RNA gene cluster, the RNA polymerase large subunit gene, and mitochondrial sequences spanning the cytochrome oxidase I gene and the mitochondrial ribosomal RNA (and intervening) gene sequences (see S1 Fig). Numbers at nodes are percentages of 1000 jackknife replicates (25% of characters randomly deleted per replicate) that supported those bipartitions. Synrhabditis and *Pellioditis* clades [11] are strongly supported. Right tree: Clann [36] supertree of bootstrapped ML trees from 7 subsets of the supermatrix (see S1 Fig). Numbers at the nodes represent the proportions of bipartitions among individual ML trees that were consistent with the branches shown. In both trees, branches are collapsed where bipartition support is less than 70% or 0.7, such that only robustly supported branches are shown. Four species groups are shown with different colors: *Pellio* group, *Papillosa* group, *Kenyensis* group and a group containing *P. huizhouensis* and *P. meridionalis*. Obligate parasitism evolved twice, once in the Angiostomatidae (represented here by *Angiostoma norvegicum*) from within *Pellioditis*, and in the lineage to *Agfa* (represented here by *Agfa flexilis*), the sister group to *Pellioditis*. (B) Character evolution traced by parsimony onto the WP phylogeny. Cupola-shaped female tails have evolved four times independently from conical-shaped tails (red lineages), and possibly again in the "slow" isolate of *P. akhaldaba*, not shown. Hermaphroditism evolved twice: in the lineages to *P. hermaphrodita* and *P. californica*.

- *Agfa* is the sister taxon of a robustly supported clade that includes *Pellioditis* + Angiostomatidae (represented by *Angiostoma norvegicum*, but other *Angiostoma* sequences are very similar). Based on the better-resolved WP phylogeny, Angiostomatidae likely branches off within *Pellioditis*. If this holds true, *Pellioditis* is paraphyletic with respect to this group of obligate parasites. The *Pellioditis* sister taxon of Angiostomatidae, however, is at present unresolved.

- There are three robustly supported clades within *Pellioditis*: a small group that contains only *P. huizhouensis* and *P. meridionalis*, a group of mostly South African species (*Kenyensis* species group), and a group that includes *P. papillosa*, *P. californica* and *P. pelhamensis* n. sp., among others (*Papillosa* species group). A fourth group that is well supported by WP but not by ML includes *P. pellio*, *P. hermaphrodita*, and *P. neopapillosa*, among others (tentatively called the *Pellio* group). Angiostomatidae may be sister to or derived from within this latter group.

- *P. hermaphrodita* and *P. neopapillosa* are likely sister species. It is interesting to note this species pair was suggested early on by Osche [99] as an example of "complementary" species [28]. In this scenario of sympatric speciation from a gonochoristic stemspecies, a uniparental (parthenogenetic or self-hermaphroditic) species splits from a gonochoristic sister species, with which it co-exists. In another example, a parthenogenetic variant was reported from an otherwise gonochoristic *P. pellio* population [9]. Also, the fact that *P. hermaphrodita* and *P. californica* are derived from separate clades shows that their hermaphroditic mode of reproduction arose independently (Fig 10B). Hermaphroditism and other modes of uniparental reproduction have arisen repeatedly in rhabditids, possibly facilitated by the architecture of the sex determination pathway as well as gene duplications unique to particular lineages [20, 21, 100].

- For both trees depicted in Fig 10A, a conoid female tail is plesiomorphic for *Pellioditis* (Fig 10B). Parsimony reconstruction suggests that cupola-shaped tails originated at least four times. Thus, this morphological feature appears to be quite homoplasious in this group.

- *P. pellio* and *P. pelhamensis* n. sp. are in separate clades, suggesting that associations with earthworms arose multiple times, not just once as proposed previously [76]. An alternative interpretation, however, is that associations with both slugs and earthworms are plesiomorphic but that there is an ascertainment bias in reporting the host: i.e., investigations of only slugs will only show that associated nematodes can be found on slugs, not that they are specifically associated with slugs. *P. pelhamensis* n. sp. and *P. pellio* were both found to be associated with slugs as well as earthworms [4–6] (D. Howe & D. Denver, pers. comm.), suggesting that a broader range of possible hosts needs to be surveyed.

- *P. pellio*—the type species of *Pellioditis*—is embedded well within the *Pellioditis* (syn. *Phasmarhabditis*) clade. This placement validates the synonymization of *Phasmarhabditis* with *Pellioditis*, and the valid name of this clade is unquestionably *Pellioditis*.

- The MP phylogeny allows an estimate of the **stemspecies pattern**:

  Body cylindrical, robust, lateral field with three prominent ridges, lips in three pairs, one dorsal and two subventral, stoma triangular, buccal cavity three-edged prismatic, relatively short, stegostom to about half the stoma length, glottoid apparatus well developed, each sector with warts, pharynx with weak median bulb, gonochoristic, female amphidelphic, vulva a transverse slit in midbody, at first ovi- and later ovoviviparous, female tail conical-shaped, phasmids papilliform projecting at a point midway between anus and tail terminus, testis ventrally reflexed, male bursa well developed, peloderan, anteriorly open, nine pairs of bursal

papillae (GP), three pairs anterior of the cloaca, two adcloacal, formula v1,v2,v3/(v4,ad)/v5(v6, v7,pd)ph, GP1 (v1) on level of spiculum head, GP1 to GP3 (v1–v3) evenly spaced, GP4 (v4) and GP5 (ad) very close, GP6 (v5) separate, phasmids (ph) papilliform, spicules separate, nearly straight, like a blade, head not offset, gubernaculum spatula-shaped, following the contour of spicules to half of their length, one precloacal sensillum inconspicuous and two postcloacal sensilla conspicuous. Necromenic in gastropods (and possibly earthworms).

## Conclusion

Here, we present a description of a species, *P. pelhamensis* n. sp., which has been used in previous comparative developmental and systematics studies [3, 4, 6, 13–17, 26, 38–48, 50–52, 54–57, 98] and in investigations of its pathogenicity to gastropods and earthworms [7]. We also present a new comprehensive redescription of *P. pellio* including sequence data, which we use to include *P. pellio* in a phylogenetic framework for related species (Fig 10).

Our new phylogenetic results clearly show that *P. pellio* is embedded in a monophyletic group containing all species described as *Phasmarhabditis*. Because of the confusing history of the taxonomy of this group (see S1 File), because *P. pellio* was never included in *Phasmarhabditis*, and probably because it was originally described from earthworms and not slugs, *P. pellio* has been systematically omitted from consideration in descriptions of new "*Phasmarhabditis*" species. As a consequence, a recently described species, *P. bohemica*, is synonymous with *P. pellio*, as shown here. There is no more ambiguity about the validity of synonymizing *Phasmarhabditis* with *Pellioditis*. Since *P. pellio* is the senior name-bearing type species, *Pellioditis* is the name of the clade, and the name *Phasmarhabditis* must be retired (see also S1 File).

Fortunately, DNA sequences are easily obtained from rhabditid nematodes, allowing their use as barcodes to help identify species as well as phylogenetically informative characters that can place species within a taxon, identify sister taxa and provide a framework for evolutionary analyses. As we and others have found, the rapidly evolving ITS1 and ITS2 regions from the nuclear ribosomal RNA gene cluster are easily amplified by PCR and make excellent barcodes [22]. Nearly complete sequences for the 18S and 28S ribosomal RNA genes also provide excellent phylogenetic information for a range of phylogenetic relationships in Rhabditidae and other nematodes, often yielding phylogenetic results very consistent with whole-genome studies, although more genes tend to improve resolution [17, 101, 102]. At a minimum, we encourage researchers to sequence as much of the ribosomal RNA gene cluster as possible. Another excellent resource is the existence of living collections, such as the cryogenic archives at the *Caenorhabditis* Genetics Center, which accepts not only *Caenorhabditis* but other rhabditid species as well, including the *Pellioditis* species described here. Living strains allow researchers to compare features that cannot be easily compared with static descriptions and slides such as behavior and development [44], and allow tests for reproductive isolation and pathogenesis of potential hosts [7, 22].

According to the more highly resolved MP cladogram (Fig 10A), *Angiostoma norvegicum* appears to branch off within *Pellioditis*. Because all *Angiostoma* sequences for the genes used here are very similar and different from sequences of other species, we are confident that these data represent the Angiostomatidae clade. This cladogram thus places Angiostomatidae far apart from *Agfa*, which is the sister group of *Pellioditis*. If this phylogenetic hypothesis holds true, obligatory gastropod parasitism evolved twice independently within Rhabditidae. The closest relative of Angiostomatidae among *Pellioditis* is currently unknown, although our MP phylogeny suggests that it might be a species of the *Pellio* group. More molecular and ecological data are needed to more confidently resolve these relationships and shed light on the evolutionary path from necromeny or gastropod pathogenicity to parasitism.

We found that earthworm-associated *P. pellio* and *P. pelhamensis* n. sp. belong to two clearly distinct groups within *Pellioditis*. Thus, the exploitation of earthworms as hosts could have evolved twice independently. Given that both species were also found to be associated with gastropods, and at least *P. pelhamensis* n. sp. was shown to be pathogenic to both gastropods and earthworms, it is also possible that an association with both host taxa is an ancestral feature of the clade. Whether any of the other *Pellioditis* species can be found on earthworms is unknown since most of them were isolated in studies that specifically investigated only gastropods. Thus, a more thorough survey of soil invertebrates for the presence of *Pellioditis* nematodes is warranted to obtain a better picture of the host range and biology of these species. In this context, It is relevant that *P. pellio* [6] and *P. akhaldaba* [74] were also found in *Galleria melonella* larvae that were used as baits for entomopathogenic nematodes. Although it is most likely that this observation is due to an opportunistic colonization of a decaying corpse, it cannot be ruled out that these species can naturally infect live insect larvae.

It is desirable to have the most comprehensive knowledge possible of the ecology and biology of a species that is developed as a biological control agent before it is released. *P. hermaphrodita* has been tested for non-target effects on several invertebrate taxa and does not appear to affect the earthworms tested [103]. However, even this species is not without concern, since a recent study [4] found Nemaslug®-derived strains of *P. hermaphrodita* in California, Oregon and New Zealand, places where Nemaslug® is not licensed for biocontrol. Going forward, more data on ecology and natural or potential host ranges are needed for *Pellioditis* species.

## Supporting information

**S1 Table. Taxa and GenBank accession numbers for DNA sequences used in phylogenetic analyses.**
(XLSX)

**S1 Fig. Schematic view of the DNA alignment supermatrix and submatrices used for supertree analyses.** To conserve space, only the parsimony-informative characters (alignment positions) are depicted, with different colors depicting different nucleotide character states. Labeled boxes of different colors represent the taxon and character subsets of the full supermatrix used for individual ML bootstrap analyses. See Materials and Methods for description of datasets.
(TIF)

**S1 File. History of the *Pellioditis* and *Phasmarhabditis* taxa, and how the confusion between them arose.**
(DOCX)

**S2 File.** Stacks of focal planes of DIC images for different parts of EM434 and SB361: **S1 Video.** EM434 adult female showing posterior spermatheca, right side view. **S2 Video.** EM434 adult female stoma region, left side view. **S3 Video.** EM434 adult female tail, ventral view. **S4 Video.** EM434 adult female pharynx region, left side view. **S5 Video.** EM434 adult male stoma region, left side view. **S6 Video.** EM434 adult male tail (copulatory bursa), ventral view. **S7 Video.** EM434 adult male tail, right side view. **S8 Video.** EM434 adult male stoma region, dorsal view. **S9 Video.** EM434 spicules (lateral views) and gubernaculum (ventral view) isolated from an adult male. **S10 Video.** SB361 adult female stoma region, right side view. **S11 Video.** SB361 adult female tail, right subventral view. **S12 Video.** SB361 adult female anterior oviduct, left side view. **S13 Video.** SB361 adult female pharynx region, left side view. **S14 Video.** SB361 adult male pharynx region, ventral view. **S15 Video.** SB361 adult male stoma region, ventral view. **S16 Video.** SB361 adult male stoma region, right side view. **S17 Video.** SB361 adult male

tail, right side view. **S18 Video.** SB361 adult male tail, ventral view. **S19 Video.** SB361 spicules (lateral views) and gubernaculum (ventral view) isolated from an adult male.
(ZIP)

## Acknowledgments

The authors thank Marjolein Couvreur, University of Ghent, for scanning electron microscopic images. We thank Dana Howe and Dee Denver, Oregon State University, for communicating and allowing us to cite unpublished results. Some strains were archived at and provided by the *Caenorhabditis* Genetics Center at the University of Minnesota and the New York University Rhabditid Collection. We thank Dominick Verschelde (Curator, Ghent University Zoology Museum), Gerrit Karssen (Curator, Wageningen Nematode Collection), and Steve Nadler (Curator, Nematode Collection, University of California, Davis) for archiving type material.

## Author Contributions

**Conceptualization:** Irma Tandingan De Ley, David H. A. Fitch.

**Data curation:** Irma Tandingan De Ley, David H. A. Fitch.

**Formal analysis:** Irma Tandingan De Ley, Karin Kiontke, Walter Sudhaus, David H. A. Fitch.

**Funding acquisition:** David H. A. Fitch.

**Investigation:** Irma Tandingan De Ley, Karin Kiontke, Walter Sudhaus, David H. A. Fitch.

**Resources:** Irma Tandingan De Ley, Karin Kiontke, Wim Bert, Walter Sudhaus, David H. A. Fitch.

**Supervision:** Wim Bert.

**Validation:** David H. A. Fitch.

**Visualization:** Karin Kiontke, Wim Bert, Walter Sudhaus, David H. A. Fitch.

**Writing – original draft:** Irma Tandingan De Ley, Walter Sudhaus, David H. A. Fitch.

**Writing – review & editing:** Irma Tandingan De Ley, Karin Kiontke, Wim Bert, Walter Sudhaus, David H. A. Fitch.

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
