## [Decision Letter · Decision Letter 0]

11 Jul 2023

PONE-D-23-19233

*Pellioditis pelhami* n. sp. (Nematoda: Rhabditidae) and *Pellioditis pellio* (Schneider, 1866), earthworm associates from different subclades within *Pellioditis* (syn. *Phasmarhabditis* Andrássy, 1976)

PLOS ONE

Dear Dr. Fitch,

Thank you for submitting your manuscript to PLOS ONE. After careful consideration, we feel that it has merit but does not fully meet PLOS ONE’s publication criteria as it currently stands. Therefore, we invite you to submit a revised version of the manuscript that addresses the points raised during the review process.

We look forward to receiving your revised manuscript.

Kind regards,

Aashaq Hussain Bhat

Academic Editor

PLOS ONE

Journal Requirements:

4.  Please take this opportunity to be sure you have met all of our guidelines for new species. For proper registration of a new zoological taxon, we require two specific statements to be included in your manuscript.

a.
In the Results section, the globally unique identifier (GUID), currently in the form of a Life Science Identifier (LSID), should be listed under the new species name, for example:  

  Anochetus boltoni Fisher sp. nov. urn:lsid:zoobank.org:act:B6C072CF-1CA6-40C7-8396-534E91EF7FBB

 Another LSID for the manuscript itself should also appear within the Nomenclature statement. You will need to contact Zoobank (zoobank.org/About) to obtain a GUID (LSID). You should receive one LSID for your manuscript and a separate, unique LSID for the new species.  b.
Please also insert the following text into the Methods section, in a sub-section to be called "Nomenclatural Acts":   The electronic edition of this article conforms to the requirements of the amended International Code of Zoological Nomenclature, and hence the new names contained herein are available under that Code from the electronic edition of this article. This published work and the nomenclatural acts it contains have been registered in ZooBank, the online registration system for the ICZN. The ZooBank LSIDs (Life Science Identifiers) can be resolved and the associated information viewed through any standard web browser by appending the LSID to the prefix "" ext-link-type="uri" xlink:type="simple">http://zoobank.org/". The LSID for this publication is: urn:lsid:zoobank.org:pub: XXXXXXX. The electronic edition of this work was published in a journal with an ISSN, and has been archived and is available from the following digital repositories: PubMed Central, LOCKSS [author to insert any additional repositories]. All PLOS ONE articles are deposited in PubMed Central and LOCKSS. If your institute, or those of your co-authors, has its own repository, we recommend that you also deposit the published online article there and include the name in your article. Following a recent ruling by the International Commission on Zoological Nomenclature, electronic journals are now a valid format for publication of new zoological taxa. In order to ensure the valid publication of your new species, please be sure to include the updated version of Nomenclatural Acts (above). A complete explanation of our guidelines for publishing new species can be found on our website: http://www.plosone.org/static/guidelines#zoological.

Additional Editor Comments:

The work is presented nicely and will surely solve the ambiguities in the taxonomy of Phasmarhabditis. The authors have done the cross-hybridization studies, but the procedure used for carrying out crosses is missing. Authors should briefly describe this in methodology. Also, I noticed grammatical errors in some parts of the manuscript which authors need to carefully check. I also encourage authors to write genus name in full while describing it for the first time. The description or redescription of any species should follow usual pattern as is followed in all descriptions of a new species.

Reviewers' comments:

Reviewer's Responses to Questions

**Comments to the Author**

1. Is the manuscript technically sound, and do the data support the conclusions?

Reviewer #1: Yes

Reviewer #2: Yes

2. Has the statistical analysis been performed appropriately and rigorously? 

Reviewer #1: N/A

Reviewer #2: N/A

3. Have the authors made all data underlying the findings in their manuscript fully available?

Reviewer #1: Yes

Reviewer #2: Yes

4. Is the manuscript presented in an intelligible fashion and written in standard English?

Reviewer #1: Yes

Reviewer #2: Yes

5. Review Comments to the Author

Reviewer #1: Submitted paper is outstanding contribution to the knowledge of rhabditids. It is known that many free-living rhabditids have biotic relationships with invertebrate and in few cases with vertebrate animals. The problem of relationships between rhabditid nematodes and terrestrial mollusks is a separate phenomenon. Unfortunately, studies in this field faced in recent years some problems: not resolved relationships with other rhabditids, contradictions in the nomenclature. The study of this group started in mid-nineteen century, when the standards of the species description were completely different. The main peculiarity of the submitted paper is in combination of several modern methods of nematological examination (nucleotide sequence analysis, SEM, staining with monoclonal antibodies) with profound and extremely accurate morphological analysis. Such a combination of approaches was only possible through the joint efforts of literally all leading specialists in this field of nematology. The paper is completely ready for publication – the text is very accurately edited. The plates of line-art drawings and SEM images are very informative. I am sure that this paper upon the publication will be actively cited in other studies of invertebrate-associated nematodes. One of the reasons for such predicable interest toward this publication is in the important taxonomic decision. After years of controversy, the authors of submitted paper finally solved problem of the interrelationships between Pellioditis and Phasmarhabditis. I am sure that after the publication of this paper the use of the generic name Pellioditis will be the only possible and reasonable.

To fix this long-lasting problem the authors are proposing in-depth redescription of Pellioditis pellio, isolated from type locality. The re-description is richly illustrated with all possible methods of morphological description including SEM an DIC images. DIC-microphotographs deserve special praise. E.g. it is quite a rare occasion when inner structure of spicules and the shape of gubernaculum are so clearly seen as on Fig. 2, G. Another valuable demonstration of the power of basic morphological information is on the Fig.2 D where the formation of anterior and posterior dorsal rays of bursa is demonstrated on the image of fourth-stage juvenile.

Another trend-setting moment of the paper is in phylogenetic analysis. As leading specialists in nematode molecular phylogeny are between the authors, some novel (at least for me) approach is proposed (supertrees from several datasets, weighted parsimony etc.). Several approaches of authors are provoking special interest and invite to use the same approach, including idea to create concatenating sequences even if primary sequences were obtained from different strains. Such an action can be justified with the fact of very uneven knowledge of separate species and strains of rhabditid nematodes.

I am sure that the paper is completely ready for publication but would like only to attract authors’ attention to few minor moments.

E.g. both in description of Pellioditis pelhami n. sp. and in re-description of Pellioditis pellio (Schneider, 1866) the number of paratype and neotype, correspondingly. in the Nematode Collection at the University of California, Davis is given: “as slide ____.” Probably number will be provided from UC Davis only on later stages, but in modern shape it looks strange. To be honest the idea to mount several specimens on the same slide with only one specimen between those serving as type looks quite new for me, but authors ensure completely precise identification of paratype/neotype with indication like ‘the rightmost male is the holotype’ and in fact the slides are soon sent to the collections.

Fig. 3G – probably an explanation is needed here – to help for the reader to find lateral field. Near the vulvar opening one can find longitudinal ridges which are probably artefact of critical-point drying. The fine features of the lateral field (true lateral field) might be traced out near the margin of the image – in the overexposed portion of the body. It is obvious if compare this image 3G with e.g. Fig.7 J – the position of lateral field in the latter is quite obvious and explained by the inset image.

And finally – authors use the binomial Angiostoma norvegicus – page 32, but primary in the first description by Ross et al (Angiostoma norvegicum n. sp. (Nematoda: Angiostomatidae) a parasite of arionid slugs in Norway) the spelling was different. Probably it would be better to add citation of the publication where this changed speclling was proposed (as we can understand to comply with neutral gender?).

Reviewer #2: The manuscript contains a description of a new species. Morphological and morphometrical description is acceptable. However, probably, the specific name proposed for the species could be not correct, which should be ending in "-ensis" because it correspond with a place/locality. Other comments and suggestions are included in the MS file.

6. PLOS authors have the option to publish the peer review history of their article (what does this mean?). If published, this will include your full peer review and any attached files.

Reviewer #1: No

Reviewer #2: No

---

## [Author Response · Author response to Decision Letter 0]

20 Jul 2023

Many thanks for the very valuable reviews of our paper, PONE-D-23-19233, re-titled "Pellioditis pelhamensis n. sp. (Nematoda: Rhabditidae) and Pellioditis pellio (Schneider, 1866), earthworm associates from different subclades within Pellioditis (syn. Phasmarhabditis Andrássy, 1976)". To the best of our understanding, we have addressed all of your (Editor's) concerns, as well as those of the reviewers, as listed below:

Editor's comments

 (Note that this letter represents the Response to Reviewers requested. Also, we have uploaded a marked-up copy of the manuscript as well as an unmarked version.)

1. I believe we have formatted our paper according to the PLOS ONE style requirements.

2. All accession numbers for data and materials (e.g. type specimens) have been assigned and are included in the manuscript.

3. We have decided not to cite "data not shown". All relevant and essential data are included in the paper, in the table, text, figures, and extensive supplemental material.

4. All guidelines for descriptions of new species have been followed. We have also registered both the relevant taxa and the paper with Zoobank and include the GUID/LSIDs in the paper (Nomenclatural Acts section).

5. We have tried to be as comprehensive as possible in reporting all the information associated with the references. (Note: we have all the references available in an EndNote "traveling library", which we can easily send to you if needed. Just let me know.)

6. We have now included the protocol for testing reproductive compatibilities in the Materials and Methods section (it was included as a footnote to a table previously, but we agree with you that it is better to include in the MM section).

7. We have tried to fix all grammatical errors (although none were specified). I have to admit that I did not find many. I'm not sure if the editor meant the type of prose found in the species descriptions per se? We have followed the typical "telegraphic" style commonly used in most zoological descriptions. In the other parts of the text, however, I found very few such errors, but such errors (as far as I know) have now been corrected.

8. We have now written out genus names for species when they are mentioned for the first time.

9. As mentioned in note 7 above, we have used the typical format for nematode species descriptions (also as in previous nematode descriptions published in PLoS ONE, e.g. P. huizhouensis).

Reviewer 1

1. We have now provided the type specimen identifiers for the UC Davis collection. (We did not yet have them when we submitted the manuscript originally.)

2. Regarding the legend to Fig. 3G, we have removed the reference to the "lateral field", which we agree is not shown well in this figure. Instead, the lateral field is better shown in other panels. We therefore call attention to the lateral field in 3I and 4E-G.

3. We have fixed the typo regarding the species name Angiostoma norvegicum.

Reviewer 2

1. We have changed the species name from P. pelhami to P. pelhamensis, as per the reviewer's suggestion. This also entailed changing the name in all associated files and databases.

2. As per the comments made in the manuscript PDF, we have

 2a. fixed the typo involving P. hermaphrodita,

 2b. taken the reviewer's suggestion to emend P. kenyaensis to P. kenyensis (and cited the appropriate ICZN article justifying this emendation).

Request to publish the peer review history

YES, we agree to publishing the peer review history. I understand that the reviewers prefer to remain anonymous.

Many thanks for all the comments from editors and reviewers, which were all very helpful in improving the manuscript. Please let me know if there is anything else we should do. We are excited to see the paper in print as soon as possible.

---

## [Editor Report · Decision Letter 1]

24 Jul 2023

PONE-D-23-19233R1*Pellioditis pelhamensis* n. sp. (Nematoda: Rhabditidae) and *Pellioditis pellio* (Schneider, 1866), earthworm associates from different subclades within *Pellioditis* (syn. *Phasmarhabditis* Andrássy, 1976)PLOS ONE

Dear Dr. Fitch,

Thank you for submitting your manuscript to PLOS ONE. After careful consideration, we feel that it has merit but does not fully meet PLOS ONE’s publication criteria as it currently stands. Therefore, we invite you to submit a revised version of the manuscript that addresses the points raised during the review process.

ACADEMIC EDITOR: ulliThe authors have made most of the changes as suggested, but the typical format for nematode species descriptions is not fully met yet. I kindly request the authors to make the following changes in the description to adhere to the standard format for publication in PLOS ONE. With these changes, the nematode species description will align with the typical format required for publication in PLOS ONE.

Stoma wider and shorter in adults than in juveniles (delete is)

Pharynx corpus cylindrical, about two times as long as isthmus with slightly enlarged metacorpus narrowing into isthmus, and a bulbous postcorpus (basal bulb) longer than wide, with striated valvular apparatus; isthmus slightly longer than basal bulb (Fig 1A, 2C) (insert comma after isthmus, and delete that is)

Sperm also found in the uteri (delete is)

Uteri of mature females often filled with developing embryos (delete are).

In older females, hatched juveniles of various stages also present inside of the female uterus (delete are).

The anterior dorsal GP in position 5 (counting from the anterior) (delete is)

The posterior dorsal GP terminal, or originates at the same level as the most posterior GP (v7) (delete is and it). 

The phasmids form small ventral papillae not fully embedded in the bursal velum (delete that are).

In strain EM434, variations in the spacing of GP1, GP2 and GP3 observed.

Aberrations also found: males with abnormal numbers of rays (one having 13 GPs on one side!) and one adult male with a long (leptoderan) tail tip (delete are).

Thus, the formula (following [11, 29]), follows: (delete is)

We look forward to receiving your revised manuscript.

Kind regards,

Aashaq Hussain Bhat, Ph.D.; Postdoctoral Fellow

Academic Editor

PLOS ONE
---

## [Author Response · Author response to Decision Letter 1]

26 Jul 2023

The FINANCIAL DISCLOSURE STATEMENT has now been UPDATED to match the Funding Information (as well as the site allowed us to): Please see Author Comments for the updated statement.

The editor provided specific comments to help us edit the new species description and change it to a "typical format for nematode species descriptions" (i.e. a 'telegraphic' style). We have made all suggested edits, and thank the reviewer for clarifying what was meant by this format.

---

## [Editor Report · Decision Letter 2]

28 Jul 2023

*Pellioditis pelhamensis* n. sp. (Nematoda: Rhabditidae) and *Pellioditis pellio* (Schneider, 1866), earthworm associates from different subclades within *Pellioditis* (syn. *Phasmarhabditis* Andrássy, 1976)

PONE-D-23-19233R2

Dear Dr. David,

We’re pleased to inform you that your manuscript has been judged scientifically suitable for publication and will be formally accepted for publication once it meets all outstanding technical requirements.

Within one week, you’ll receive an e-mail detailing the required amendments. When these have been addressed, you’ll receive a formal acceptance letter, and your manuscript will be scheduled for publication.

Kind regards,

Aashaq Hussain Bhat, 

Academic Editor

PLOS ONE
---

## [Editor Report · Acceptance letter]

15 Aug 2023

PONE-D-23-19233R2 

*Pellioditis pelhamensis* n. sp. (Nematoda: Rhabditidae) and *Pellioditis pellio* (Schneider, 1866), earthworm associates from different subclades within *Pellioditis* (syn. *Phasmarhabditis* Andrássy, 1976) 

Dear Dr. Fitch:

I'm pleased to inform you that your manuscript has been deemed suitable for publication in PLOS ONE. Congratulations! Your manuscript is now with our production department. 

Kind regards, 

on behalf of

Dr. Aashaq Hussain Bhat 

Academic Editor

PLOS ONE